# Enhancing energetic disorder in all-organic composite dielectrics for high-temperature capacitive energy storage

Tan Zeng[1,3], Li Meng[2,3], Qiao Li[1] ✉, Dongduan Liu[1], Qian Zhou[1], Jinliang He [2], Qi Li [2] ✉ & Chao Yuan [1] ✉

The urgent demand for capacitive energy storage at elevated temperatures is limited by significant leakage currents in existing polymer dielectrics, which lead to excessive heat generation and increase the risk of thermal runaway. Here we demonstrate a strategy to mitigate conduction loss by modulating energetic disorder within the polymer matrix. Incorporation of high-polarity organic molecules into polyetherimide enhances dipole-dipole interactions, increasing energetic disorder and thereby decreasing charge carrier mobility. Experimental measurements and computational simulations reveal that disorder-induced energy fluctuations broaden the energy separation between transport states, effectively suppressing charge transport. The resulting composite delivers an energy density of 6.45 J cm$^{-3}$ with a charge-discharge efficiency of 90% at 200 °C, and exhibits stable performance over 100,000 cycles under an applied field of 400 MV m$^{-1}$. The observed uniformity and quality of the all-organic composite films address the challenges of scalable manufacturing for dielectric films, offering a practical pathway for the development of high-temperature dielectric materials.

Dielectric capacitors, characterized by ultrahigh power density and fast charge-discharge rates, are extensively utilized as energy storage devices in advanced electronic and power systems[1–3]. Polymer dielectrics are preferred for high-energy-density capacitors due to their high breakdown strength and easy processing[4–6]. The most commonly available commercial dielectric polymer, biaxially oriented polypropylene (BOPP), is limited to operating temperatures below 105 °C, beyond which the temperature and electric field-dependent conduction losses in the polymer dielectric can generate significant Joule heat, potentially resulting in thermal runaway in the capacitor[7–9]. Recently, the surge in demand for capacitors capable of functioning at elevated temperatures (≥150 °C) in compact energy storage systems such as electric vehicles, renewable energy management, electromagnetic pulse weapons, and aerospace power systems has imposed stringent performance requirements on polymer dielectrics[10–12]. Various

engineering polymers with high glass transition temperatures ($T_g$) such as polystyrene (PS), polyethylene naphthalate (PEN), polyimide (PI), poly(ether ether ketone) (PEEK), fluorene polyester (FPE), and polycarbonate (PC) have been proposed as high-temperature dielectrics to meet critical demands[13–16]. These dielectric polymers with high thermal and mechanical stability regarding chemical structure rely on aromatic or heteroaromatic backbones[17–19]. Unfortunately, the conjugated planar segments may inevitably compromise the capacitive performance of the polymer dielectrics under high temperatures and electric fields because of the significant conduction loss caused by the delocalization of electrons over the cyclic structure[20,21].

Inhibiting charge transport to minimize conduction loss has become a primary focus in designing high-temperature polymer dielectrics[22,23]. One practical approach involves incorporating high-insulating inorganic components, such as Al$_2$O$_3$, HfO$_2$, BNNS, SiO$_2$, and

[1]College of Electrical and Information Engineering, Hunan University, Changsha, Hunan, China. [2]State Key Laboratory of Power System, Department of Electrical Engineering, Tsinghua University, Beijing, China. [3]These authors contributed equally: Tan Zeng, Li Meng. ✉e-mail: qiaoli@hnu.edu.cn; qili1020@tsinghua.edu.cn; chaoyyy@outlook.com

MgO, into the polymer matrix to create charge traps or depositing inorganic layers with wide band gap on the polymer surface to form charge-injecting barriers[24–27]. However, phase separation and poor flexibility of these composite polymers pose challenges for large-scale film preparation[28–30]. Another all-organic strategy includes introducing high-electron-affinity molecules, such as PCBM, ITIC-Cl, TE, and NTDCA into aromatic polymers to trap the charge carriers[31–33]. Furthermore, ladderphane copolymers with high-electron-affinity units or polymers with nonplanar structures and non-conjugated backbones are developed to suppress conduction loss[9]. Nonetheless, it remains uncertain whether the energy levels of organic structure-based traps are adequate to prevent charge detrapping under high electric fields[34,35].

Most aromatic polymers are amorphous, lacking long-range molecular order, with charge transport occurring via a field-assisted, thermally activated hopping mechanism intrinsically linked to the degree of energetic disorder[36–38]. Energetic disorder is defined as the Gaussian distributed scattering of electronic transport energy levels within the polymer, arising from the interplay between polymer conformation and distribution of electronic energy levels. Structurally, it is dictated by the spatial and conformational dynamics of the polymer matrix, whereas electronically, it stems from disruptions in the distribution of electronic energy levels[39]. Figure 1a presents a schematic diagram illustrating the effect of charge transport by energetic disorder. Enhanced energetic disorder broadens the distribution of energy levels, lowering carrier hopping probability and increasing the thermal activation energy required for charge mobility[37,40,41].

Conversely, a narrower energy distribution promotes nearly equivalent energy sites, diminishing reliance on thermal assistance[42].

Here, departing from the previous approaches, we propose a method to modulate the energetic disorder to suppress the charge transport by comprising high-polarity organic molecules into the polymer matrix. The fundamental principle involves an extended energy difference between transport states resulting from dipole-dipole interactions, enhancing energetic disorder in the composites. The key findings are that enhanced energetic disorder significantly inhibits charge carrier dynamics, crucial for minimizing energy losses and leakage currents. Therefore, the resultant all-organic composites demonstrate an electrical conductivity two orders of magnitude lower than that of pristine polymer under elevated temperatures. This pronounced decrease in conductivity contributes to enhanced capacitive performance at elevated temperatures, as reflected by the increased values of $U_d$ and $\eta$. These all-organic composites can be easily fabricated at scale while maintaining high quality.

## Results and discussions

### Modulation of energetic disorder

Polyetherimide (PEI), a promising candidate for high-temperature electrostatic energy storage, is chosen as the polymer matrix due to its demonstrated thermal stability and mechanical strength (Supplementary Fig. S1). Energetic disorder arises from the variations in the local energy landscape, influenced by intermolecular forces such as van der Waals interactions, hydrogen bonds, and dipole-dipole

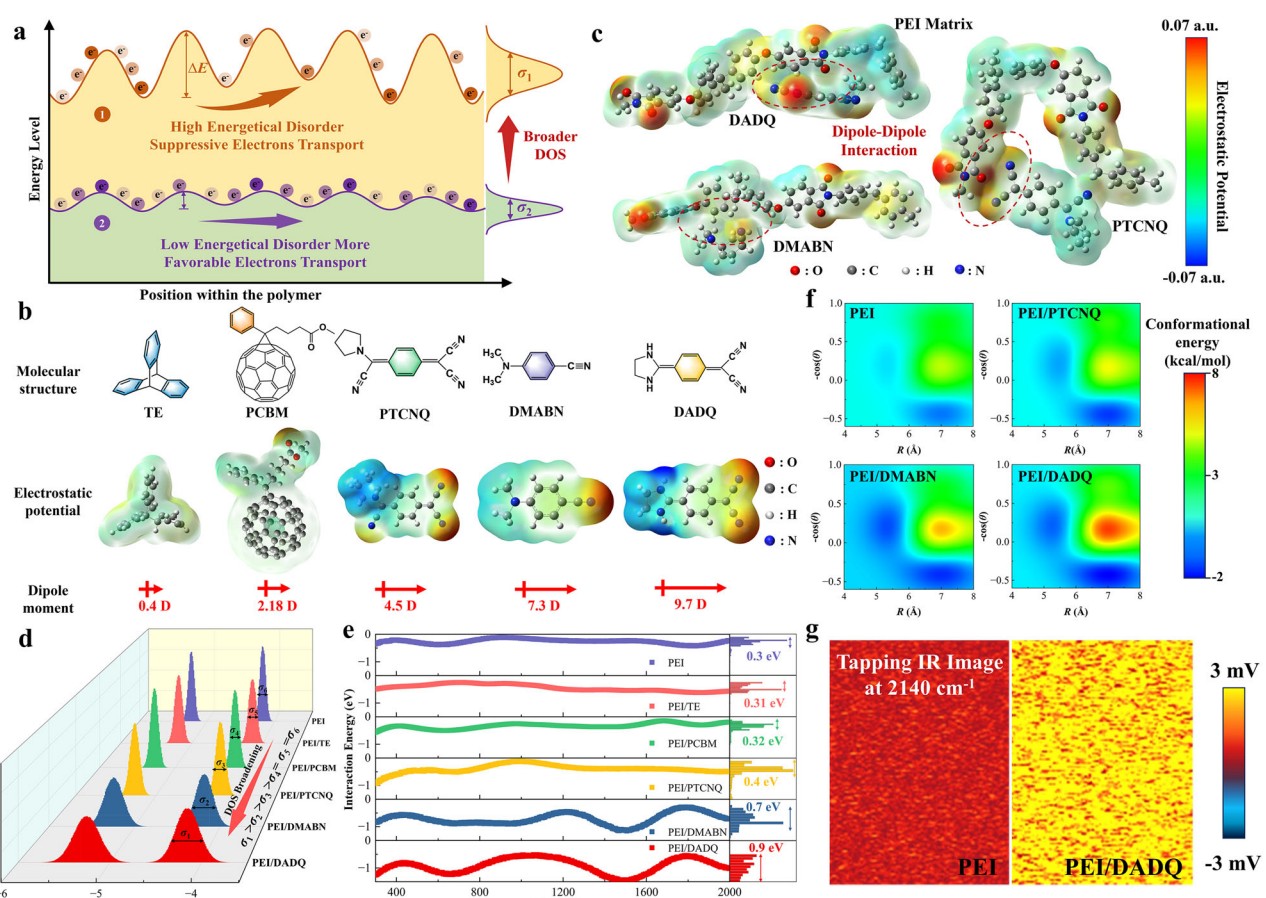

**Fig. 1 | Modulation of energetic disorder by dipole-dipole interaction.** **a** Schematic diagram illustrating the effect of charge transport by energetic disorder. **b** Molecular structures, electrostatic static potential (ESP) surfaces and dipole moments of TE, PCBM, PTCNQ, DMABN and DADQ. **c** Depiction of dipole-dipole interactions between high-polarity organic molecules (PTCNQ, DMABN,

DADQ) and polymer chain. **d**, **e** Analysis of distribution of electronic energy levels and interaction energy fluctuations for pristine PEI, PEI/TE, PEI/PCBM, PEI/PTCNQ, PEI/DMABN and PEI/DADQ composites. **f** Analysis of conformational energy distribution for pristine PEI, PEI/PTCNQ, PEI/DMABN and PEI/DADQ composites. **g** Nano-IR mapping at 2140 cm⁻¹ of pristine PEI and PEI/DADQ composite.

interactions[43,44]. The local energy landscape is characterized as a collective outcome of intermolecular and intramolecular interactions within the polymer composite[45]. Notably, the relatively strong dipole–dipole interactions contribute to structural stabilization, particularly at elevated temperatures[46]. These interactions can also be effectively tuned by introducing appropriate dipolar dopants, allowing for adjustments in energetic disorder without compromising the fundamental mechanical and thermal properties of the polymer matrix[47–49]. Previous studies on all-organic composite dielectrics commonly incorporated high electron affinity molecules, such as triptycene (TE) and [6,6]-phenyl-$C_{61}$-butyric acid methyl ester (PCBM)[10,16]. However, as shown in Fig. 1b, these molecules exhibit low dipole moments of 0.4 D and 2.18 D, respectively, attributed to their highly symmetric molecular structures. In contrast, we propose incorporating high-polarity molecules constructed by linking donor and acceptor groups via a π-conjugated bridge, such as 7,7,8,8-tetracyanoquinodimethane (PTCNQ), 4-(dimethylamino) benzonitrile (DMABN), and 2,5-Diamino-3,6-dicyano-1,4-benzoquinone (DADQ), with dipole moments of 4.5 D, 7.3 D, and 9.7 D, respectively, into PEI to modulate energetic disorder. Figure 1c presents the electrostatic potential distribution of the PEI/PTCNQ, PEI/DMABN, and PEI/DADQ composites, revealing pronounced intermolecular interactions between the high-polarity molecules and the PEI chains. These interactions contribute to localized variations in the electrostatic environment, highlighting the influence of dopant incorporation on the polymer matrix.

To elucidate the effect of dipole-dipole interactions on the energetic disorder in the polymer matrix, we calculated the distribution of electronic energy levels and interaction energy distribution of PEI, PEI/TE, PEI/PCBM, PEI/PTCNQ, PEI/DMABN and PEI/DADQ composites using molecular dynamics (MD) simulations (Fig. 1d, e). Doping with low-polarity molecules TE and PCBM induced minimal changes in the distribution of electronic energy levels and energy fluctuations, closely aligning with pristine PEI. Nevertheless, incorporating high-polarity molecules PTCNQ, DMABN, and DADQ, with dipole moments exceeding 4.0 D, led to a pronounced broadening of the electronic energy level distribution and heightened energy fluctuations. These findings suggest that the enhanced dipole–dipole interactions introduced by these dopants significantly modulate the electronic state landscape of the polymer matrix. Notably, PEI/DADQ composite exhibited the most pronounced broadening of electronic energy levels, underscoring the substantial impact of dipole–dipole interactions in amplifying energetic disorder and increasing variability within the transport states. The interaction energy distribution shows fluctuations within the range of 0.9 eV for PEI/DADQ, 0.7 eV for PEI/DMABN, 0.4 eV for PEI/PTCNQ and 0.3 eV for PEI. This pronounced fluctuation correlates with reduced charge hopping rates, as confirmed by hopping conduction analyses discussed later. Furthermore, as shown in Fig. 1f, these high-polarity molecules are employed to construct energy-minimized conformation models with PEI chains. All composites exhibited a more significant conformational energy difference than PEI, indicating an increase in accessible conformational states and enhanced configurational entropy, which reflects a higher number of possible arrangements and increased disorder.

Since the energetic disorder is easily influenced by the dispersibility of the doping organic molecules, we present direct chemical maps of PEI/DADQ and pristine PEI using nanoscale infrared spectroscopy (Nano-IR), employing an AFM tip to detect the sample's thermal expansion caused by infrared radiation locally. The photothermal expansion-based Nano-IR approach employed in this study achieves a spatial resolution of approximately 10 nm, offering remarkable surface sensitivity. As demonstrated in the Nano-IR images of the polymers (Fig. 1g), the notable difference is the detection of absorption at $2140\,cm^{-1}$ (C≡N stretch) at the PEI/DADQ compared to pristine PEI, indicating that the DADQ molecule exhibits uniform dispersion within the polymer matrix. This observed compatibility can be attributable to strong dipole-dipole interactions between the high-polarity molecules and the polymer. FTIR spectra of the composites indicate no significant chemical interaction between the incorporated high-polarity molecules and the polymer (Supplementary Fig. S2). Notably, the glass transition temperature ($T_g$) of PEI and its composites remains above 200 °C (Supplementary Fig. S4). The temperatures corresponding to a 5% weight loss for PEI and its composites were around 490 °C (Supplementary Fig. S5), enabling the composites to achieve favorable dielectric performance at elevated temperatures. Mechanical testing demonstrates that high-performance polymers, including PEI and its composites with high-polarity small molecules such as DADQ, DMABN, and PTCNQ, exhibit mechanical strengths similar to that of pristine PEI, as shown in Supplementary Fig. S7. These results suggest that the incorporation of the organic dopants does not significantly affect the intrinsic mechanical properties of the PEI matrix.

## Characterization of charge carrier dynamics

To verify the enhancement of energetic disorder in polymers induced by doping with high-polarity organic molecules, steady-state photoluminescence (PL) mapping and Kelvin probe force microscopy (KPFM) were utilized. As shown in Fig. 2a, we compared the wide-field PL intensity images of PEI/DADQ, PEI/DMABN, PEI/PTCNQ, and pristine PEI, recorded at 390 nm. Spatial energetic disorder in the films is evidenced by variations in the PL spectrum, resulting in different spatial distributions of PL intensity at the same wavelength. Whereas the PL intensity images of pristine PEI display uniform spatial distributions at 390 nm, the PEI/DADQ, PEI/DMABN, and PEI/PTCNQ composites show distinct and varied spatial distributions at this wavelength. This observation suggests that PEI/DADQ has the most significant spatial energetic disorder compared to PEI/DMABN, PEI/PTCNQ, and pristine PEI. To quantify the underlying energetic disorder within the composites, we computed the PL intensity and generated corresponding histograms from the spatial maps, as shown in Fig. 2b. The histograms for pristine PEI display a narrow distribution with a σ of 0.052. In contrast, the histograms for the PEI/DADQ, PEI/DMABN, and PEI/PTCNQ composites reveal significantly broader distributions, with standard deviation σ values of 0.259, 0.137 and 0.076, respectively. These findings indicate an increase in spatial energetic disorder in the composites, progressing from PEI/PTCNQ to PEI/DMABN to PEI/DADQ. Given the corresponding decrease in dipole moments, we propose that enhanced dipole-dipole interactions contribute to the elevated spatial energetic disorder observed in these polymer composites.

While PL mapping provides insight into the overall increase in energetic disorder induced by high-polarity molecules doping, its spatial resolution is confined to the micron scale. To achieve a more localized understanding, KPFM was employed to investigate nanoscale surface potential variations, providing a complementary perspective on the effects of organic molecules doping on the polymer's energetic disorder. We mapped the surface potential of PEI/DADQ, PEI/DMABN, PEI/PTCNQ, and pristine PEI (Fig. 2c), where the surface potential distribution of pristine PEI is shown to be relatively uniform. Conversely, when high-polarity organic molecules such as DADQ, DMABN, and PTCNQ are doped into the PEI matrix, the surface potential distribution reveals significant variations and broader distributions. These changes are primarily attributed to the introduction of strong dipole-dipole interactions, which disrupt the electronic could distribution along the polymer chains, inducing greater energy differences between transport states. To quantify the underlying energetic disorder within the composites, we depicted the surface potential spatial distribution of all films by a histogram. The histograms for the PEI film display a narrow distribution with a standard deviation σ of 4.1 mV. In contrast, the histograms for PEI/DADQ, PEI/DMABN, and PEI/PTCNQ exhibit much broader distributions with σ values of 12.9, 10.1, and 8.0 mV, respectively. These measurements reveal a progressive

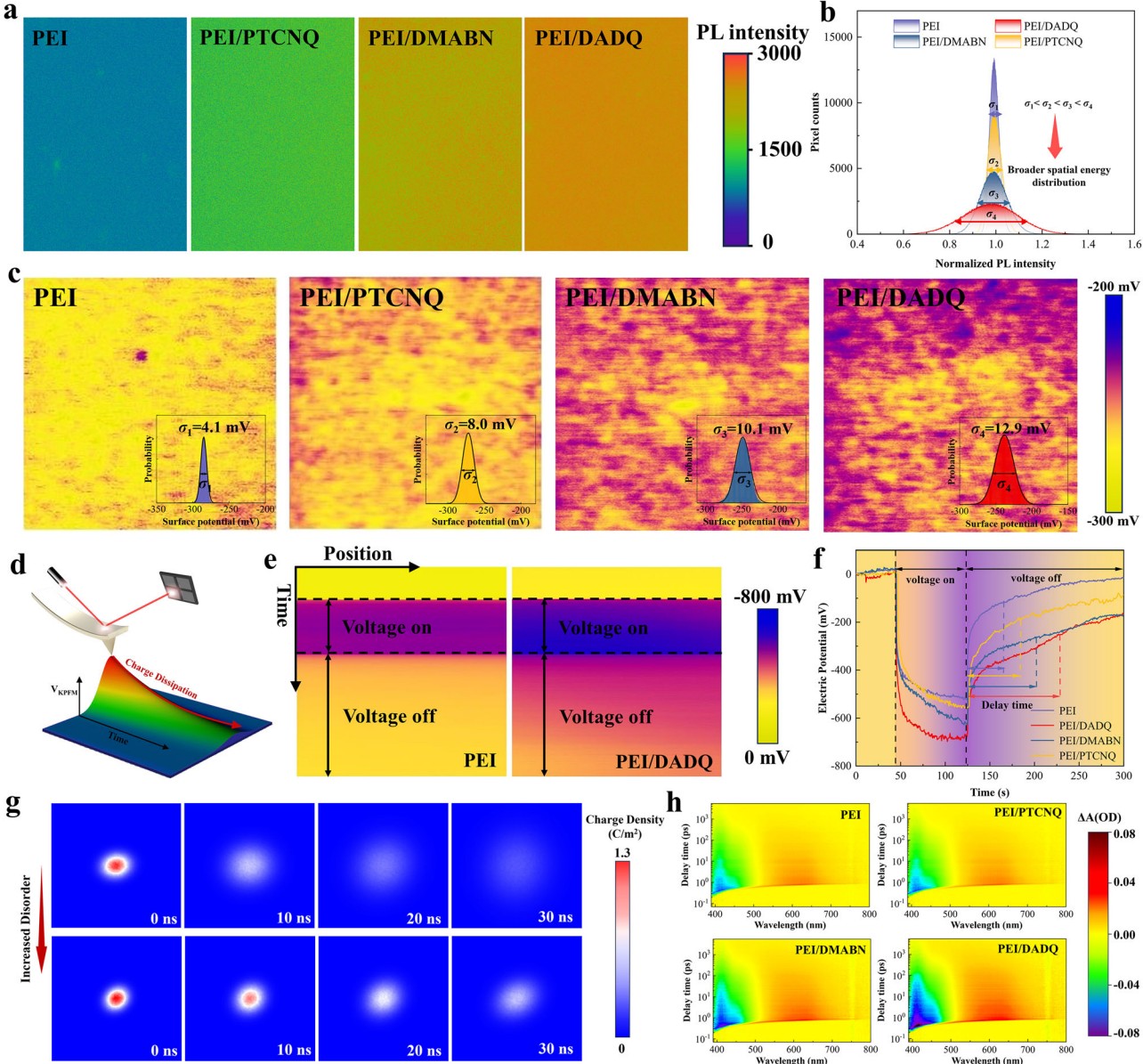

**Fig. 2 | Suppression of charge carrier transport via enhanced energetic disorder. a** Wide-field PL images of PEI/DADQ, PEI/DMABN, PEI/PTCNQ and pristine PEI under excitation at 390 nm. **b** Histograms of the PL intensity ratio, illustrating the distribution of PL intensity ratios for PEI/DADQ, PEI/DMABN, PEI/PTCNQ and pristine PEI, are shown. The $\sigma$ values represent the width of each histogram and provide an estimate of the energetic disorder. **c** Surface potential mapping of PEI/DADQ, PEI/DMABN, PEI/PTCNQ and pristine PEI, with the histogram of measured surface potential and its corresponding Gaussian fit shown in the bottom right corner for the composites. **d** Schematic diagram of the Nano-ISPD measurements conducted in KPFM mode. **e** Mapped time-dependent surface potential distribution on the scanned line in PEI/DADQ and pristine PEI. **f** Time-dependent surface potential decay in PEI/DADQ, PEI/DMABN, PEI/PTCNQ, and pristine PEI. **g** Simulated 2D charge diffusion images of polymer films with varying energetic disorders at time intervals of 0, 10, 20, and 30 ns. **h** Femtosecond transient absorption microscopy of PEI/DADQ, PEI/DMABN, PEI/PTCNQ, and pristine PEI pumped using a 390 nm laser.

increase in spatial energetic disorder from pristine PEI to PEI/PTCNQ, PEI/DMABN, and finally to PEI/DADQ. Therefore, the broader PL distribution, which indicates greater availability of states for charge carriers, combined with increased surface potential variations resulting from energy fluctuations, provides compelling evidence that dipole-dipole interactions significantly enhance the energetic disorder in polymer composites.

The charge carriers in polymers can be classified into injected and excited carriers. To further characterize a nanoscale spatially evolving for injected carriers, we utilized the nano-isothermal surface potential decay measurement (Nano-ISPD) method within KPFM to measure the surface potential on PEI/DADQ, PEI/DMABN, PEI/PTCNQ and pristine PEI after voltage on and voltage off (Fig. 2d, e and Supplementary

Characterization). The results reveal that more injected carriers accumulated on PEI/DADQ, PEI/DMABN, PEI/PTCNQ composites and dissipated more slowly than that for the pristine PEI. To quantitatively assess the process, the time-dependent surface potential decay at the midpoint of the scanned line was recorded (Fig. 2f). The initial surface potential measured on the PEI/DADQ, PEI/DMABN, PEI/PTCNQ film were − 718.8 mV, − 674.2 mV, − 609.8 mV respectively, which are significantly higher than that of the pristine PEI film (− 547.3 mV) after 60 s of charging. In addition, the potential decay rate of the pristine PEI film was faster than that of the composite films. The time required for the surface potential to drop to 50% of its initial value is 38.5, 26.9, 21.4 and 4.6 s for the PEI/DADQ, PEI/DMABN, PEI/PTCNQ film and pristine PEI, respectively. Moreover, to elucidate the impact of energetic disorder

on charge carrier transport, we conducted simulations of the dynamic process of charge carrier diffusion in polymers with varying degrees of energetic disorder (Fig. 2g and Supplementary Data 1). Initially, charge carriers are injected at the center of polymers under identical conditions. These injected charges subsequently diffused symmetrically in different in-plane directions, with a reduction in charge density. A comparison of the spatial and temporal distribution of charge density reveals that charge carriers diffuse more rapidly and uniformly in polymers with low energetic disorder. Conversely, polymers with high energetic disorder exhibit a more concentrated charge distribution and slower dissipation of carriers, attributed to the decrease in carrier mobility with increasing energy disorder, which inhibits charge carrier transport. Both experimental and simulated results verified that enhanced energetic disorder corresponds to inhibiting charge carriers dynamic in the polymers. To gain deeper insight into the structural organization of pristine PEI and PEI/DADQ, we conducted small-angle X-ray scattering (SAXS) and wide-angle X-ray diffraction (WAXD) analyses (Supplementary Figs. S18–S21). The WAXD patterns for both pristine PEI and PEI/DADQ display isotropic diffraction signals, reinforcing their predominantly amorphous nature and indicating the absence of long-range molecular ordering. Notably, the incorporation of DADQ induces subtle structural perturbations, evidenced by a slight increase in the mean intermolecular distance from 5.1 Å in pristine PEI to 5.3 Å in PEI/DADQ, alongside a reduction in coherence length from 10.6 Å to 9.8 Å. This decrease in short-range ordering suggests that DADQ doping disrupts molecular packing, leading to an increase in energetic disorder within the polymer matrix. Meanwhile, the SAXS patterns exhibit no discernible scattering features, with the signal gradually attenuating from the center outward. Consistently, the one-dimensional SAXS profiles of both pristine PEI and PEI/DADQ thin films show an absence of distinct scattering peaks, indicating a uniform electron density distribution without evidence of periodic layered structures.

To investigate the excited carriers dynamic influenced by energetic disorder on ultrafast timescales, ranging from femtoseconds to nanoseconds, we performed femtosecond transient absorption (TA) spectroscopy and time-resolved photoluminescence (TRPL) mapping on the polymer films. TA spectroscopy serves as a valuable tool for capturing both the decay of excited states and the evolution of intermediate states across timescales spanning femtoseconds to microseconds, helping to elucidate how energetic disorder influences the initial excitation and relaxation pathways. The polymer films (PEI/DADQ, PEI/DMABN, PEI/PTCNQ, and pristine PEI) were excited using 390 nm femtosecond pump pulses. The differential absorbance ($\Delta A$) between the pumped and unpumped states of the composites was measured across the 400–800 nm spectral range utilizing a femtosecond visible continuum probe. The two-dimensional variation of $\Delta A$ as a function of probe delay time and wavelength for these samples is depicted in Fig. 2h, illustrating the correlation between the excited carriers dynamics and energetic disorder in the composites. All polymer composites exhibit a more complex and extended $\Delta A$ landscape over time compared to the simpler and more rapid decay seen in pristine PEI. This complexity arises from the increased energetic disorder, where the interactions between high dipole moment molecules with the polymer chain create numerous availability of states for carriers, resulting in carriers experiencing a complex potential energy landscape and variable decay pathways. Consequently, the increased energetic disorder within the composites significantly inhibits excited carrier transport, as evidenced by the slower dissipation dynamics and extended $\Delta A$ signals.

To determine excited carrier diffusion behavior over nanosecond timescales, particularly at elevated temperatures, we compared TRPL measurements on the PEI/DADQ and pristine PEI films (Fig. 3a and Supplementary Characterization). The PL intensity images of pristine PEI and PEI/DADQ at various delay times and 200 °C are shown in Fig. 3b. The PL intensity, which is directly proportional to the carrier population, provides insight into the temporal evolution of the carrier distribution induced by the excitation laser beam. As the delay time increases, the observed expansion of the PL spot diameter indicates the diffusion of carriers away from the initial excitation site. The carrier transport process is further illustrated in the normalized 1D profiles of the PL intensity cross-section, which show broadening distribution peaks with increasing delay time (Fig. 3c). The temporal evolution of the PL image and distribution profile for PEI/DADQ reveals suppression of excited charge transport compared to pristine PEI, consistent with earlier observations (Fig. 2f). This suppression is attributed to the broadened distribution of electronic energy levels and increased energetic disorder arising from dipole-dipole interactions. The thermally stimulated depolarization current (TSDC) measurements were performed on PEI/DADQ and pristine PEI to investigate the effect of dipole-dipole interactions on localized states. As indicated in Fig. 3d, the peak of the depolarization current, indicative of carrier detrapping, is observed at a lower temperature in pristine PEI (161.9 °C) compared to PEI/DADQ (184.2 °C). This verifies that the formation of more available energy states and increased energy difference between transport states is induced by dipole-dipole interactions, which can immobilize charge carriers and prevent them from contributing to electrical conduction.

Figure 3e and Supplementary Fig. S22 illustrate the relationship between leakage current density and applied electric field at 200 °C for composites with varying concentrations of high-polarity organic molecules. At an optimal doping level of 0.5 vol%, all-organic composites (PEI/DADQ, PEI/DMABN, PEI/PTCNQ) exhibit minimal leakage current. The $J$-$E$ behavior aligns with the hopping conduction model ($R^2 = 0.984$-$0.99$), indicating that hopping conduction is the dominant conduction mechanism under elevated temperature and electric field. Notably, the minimum hopping distance for PEI/DADQ is reduced to 0.88 nm from 1.79 nm in pristine PEI, reflecting how enhanced energetic disorder effectively reduces carrier hopping distance. Dielectric properties, specifically $\varepsilon_r$ and dissipation factor, were evaluated across temperature and frequency ranges (Supplementary Figs. S25–S27). Minor variations in $\varepsilon_r$ and dissipation factor were observed across frequencies from $10^1$ to $10^6$ Hz at 150 and 200 °C, attributed to the rigidity of the PEI backbone. Temperature-dependent dielectric spectra from 30 to 200 °C (Supplementary Fig. S28) further confirm robust dielectric stability below $T_g$, underscoring suitability for capacitor applications. Weibull analysis reveals that the $E_b$ of PEI composites is significantly enhanced by doping with high-polarity molecules, particularly at low loadings and across varied temperatures (Fig. 3f, g and Supplementary Figs. S29–S32). PEI/DADQ achieves the highest $E_b$, reaching 756.5 MV/m at 150 °C and 688.3 MV/m at 200 °C, outperforming other composites. Phase-field simulations further indicate that the increased energetic disorder in PEI/DADQ effectively restricts breakdown progression, underscoring the pivotal role of energetic disorder in advancing both breakdown strength and energy storage performance (Fig. 3h).

## Capacitive energy storage performance

The energy storage capabilities at temperatures above 150 °C are critical for applications in harsh environments. The capacitive performance of PEI composites at 150 °C and 200 °C was assessed using the unipolar electric displacement-electric field (D-E) loops, as detailed in Supplementary Figs. S37–S40. These D-E loops illustrate the relationship between electric displacement and electric field strength in dielectric materials under a unidirectional electric field, depicting the charge displacement within the material during the application and removal of the electric field. In PEI composites doped with high-polarity organic molecules, the observed slimmer D-E loops with reduced remnant polarization indicate a significant reduction in energy loss compared to pristine PEI. As summarized in Supplementary Figs. S41–S43, the capacitive energy storage performance of the

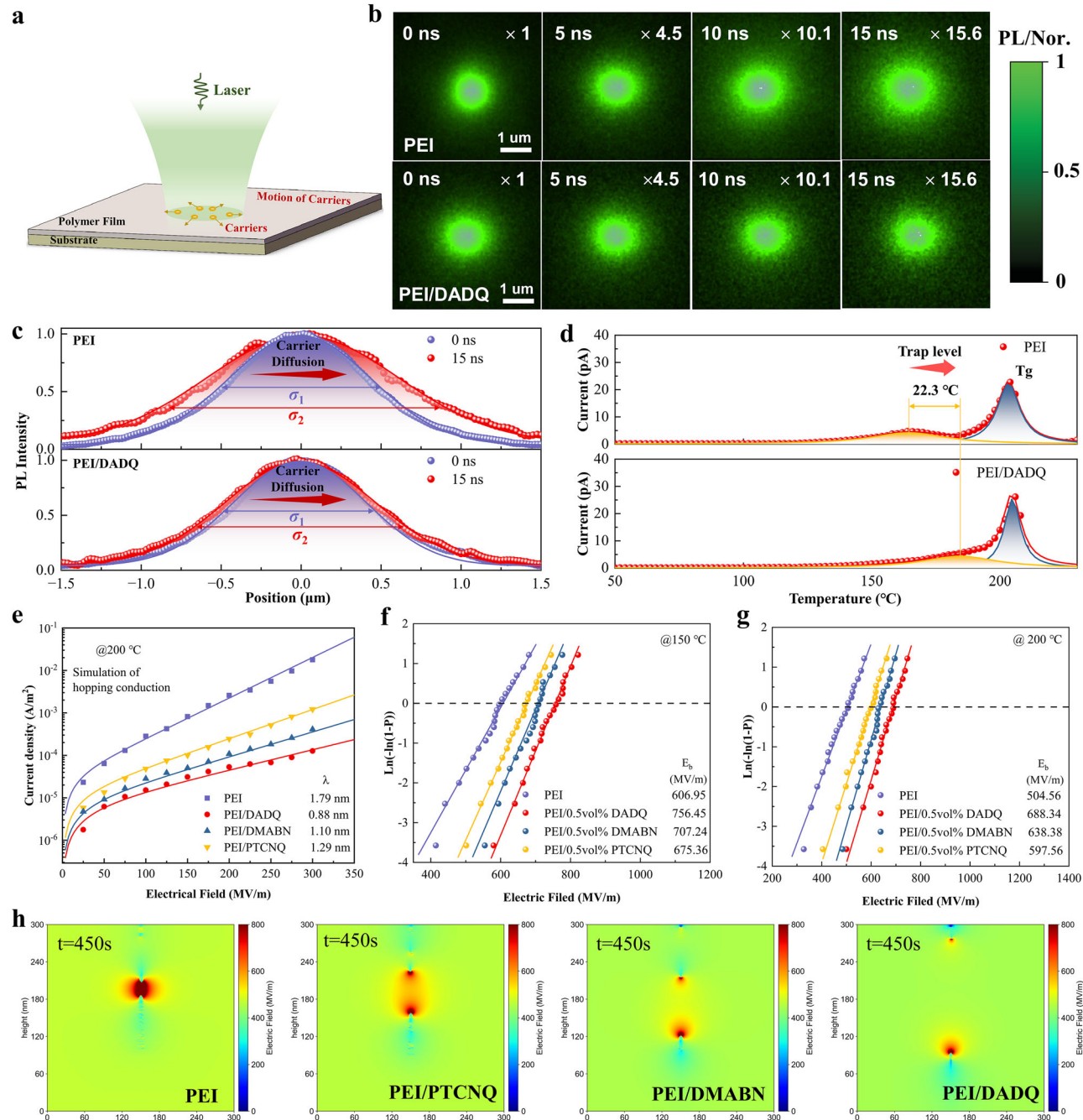

**Fig. 3 | Characterization of charge carrier dynamic and dielectric strength.** **a** Schematic representation of TRPL illustrating excited carriers transport in the composites. **b** PL intensity image of charge distribution in pristine PEI and PEI/DADQ at different delay times and 200 °C. **c** Cross-sectional profiles of the PL intensity images of pristine PEI and PEI/DADQ along the x-axis, fitted with Gaussian functions at 0 ns and 15 ns, at a temperature of 200 °C. The maximum PL signal has been normalized for comparison. **d** TSDC curves of the PEI/DADQ and pristine PEI. **e** Current density of PEI/DADQ, PEI/DMABN, PEI/PTCNQ and pristine PEI as a function of the electric field at 200 °C. Two-parameter Weibull distribution analysis of the breakdown strength of PEI/DADQ, PEI/DMABN, PEI/PTCNQ and pristine PEI at (**f**) 150 °C and (**g**) 200 °C. **h** Electrical breakdown phase evolution in PEI/DADQ, PEI/DMABN, PEI/PTCNQ and pristine PEI.

composite films, including PEI/DADQ, PEI/DMABN, PEI/PTCNQ and pristine PEI, each with a thickness of approximately 10 μm, was measured at various temperatures and dopant concentrations to assess their potential for high-temperature applications. As shown in Fig. 4a, c, the composites demonstrate a notable enhancement in $U_e$ and $\eta$ in comparison to those of pristine PEI, which has a $U_e$ of 3.79 J cm⁻³ at 150 °C and 2.75 J cm⁻³ at 200 °C. Notably, the PEI/DADQ displays the highest capacitive performance among the composites, achieving a $U_e$ of 7.19 J cm⁻³ with an $\eta$ of 93.9% at 700 MV m⁻¹ and 150 °C, and a $U_e$ of 6.45 J cm⁻³ with an $\eta$ of 90% at 650 MV m⁻¹ and

200 °C. In addition, the PEI/DMABN and PEI/PTCNQ composites also show enhanced performance compared to pristine PEI, with $U_e$ values exceeding 5.45 J cm⁻³ at 150 °C and 4.07 J cm⁻³ at 200 °C, both with efficiencies above 90%. The observed differences in $U_e$ align with variations in $E_b$ and electric conductivity, further validating the effectiveness of doping high-polarity organic molecules to enhance the capacitive properties of the polymer matrix.

Figures 4b and d compared the $U_e$ at 150 °C and 200 °C, with $\eta \geq 90\%$, between PEI/DADQ and those reported in the literature. Evidently, PEI/DADQ, with the $U_e$ of 6.45 J cm⁻³ with $\eta$ of 90%, outperforms

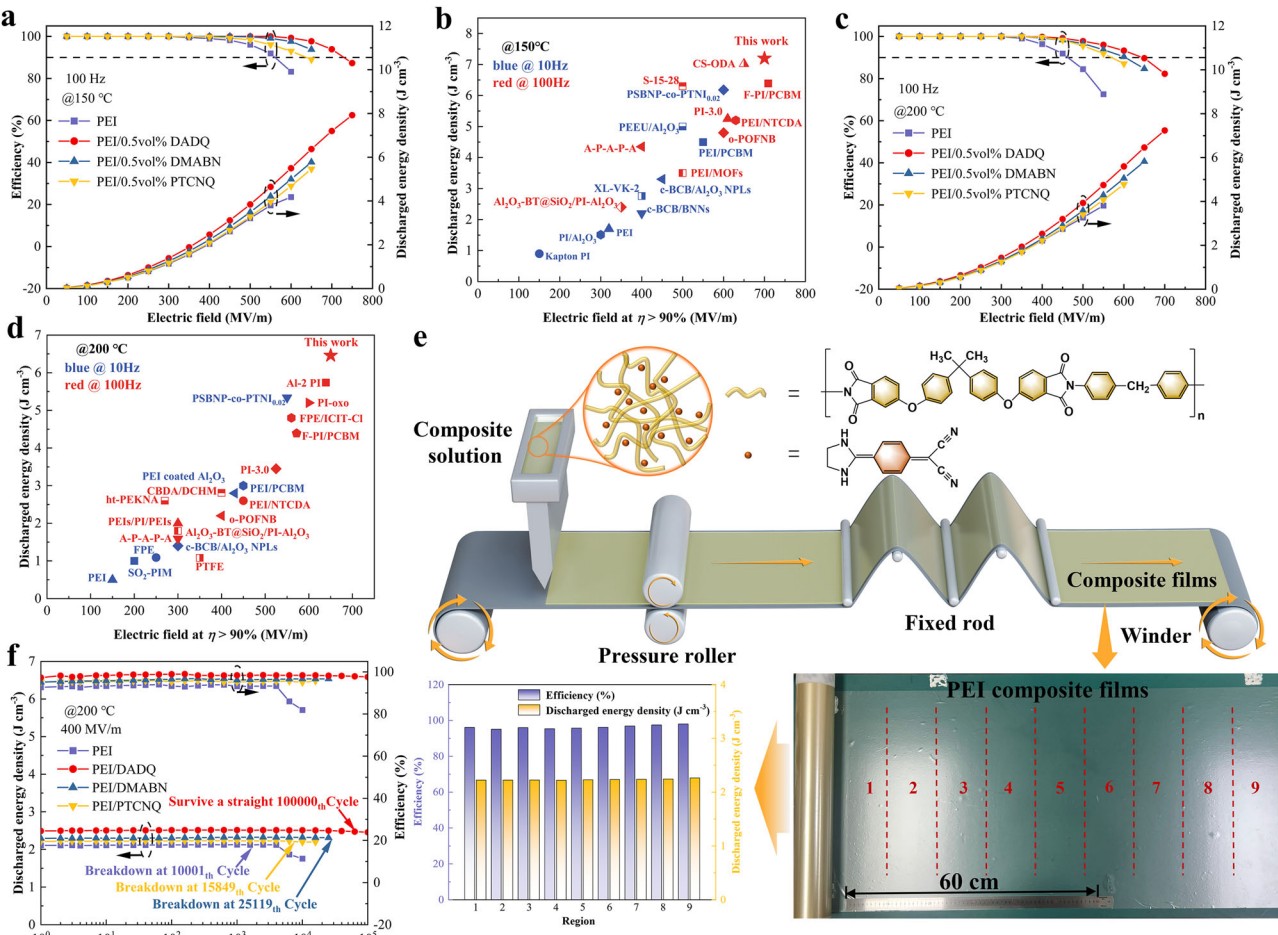

**Fig. 4 | Capacitive energy storage performance and large-area all-organic composite film fabrication.** Discharged energy density and efficiency of PEI/DADQ, PEI/DMABN, PEI/PTCNQ and pristine PEI at (**a**) 150 °C and (**c**) 200 °C and 100 Hz. Comparison of the discharged energy density at above 90% efficiency of PEI/DADQ and currently published polymer composites at (**b**)150 °C and (**d**)200 °C.

**e** Schematic representation of solution-casting production for PEI/DADQ films, accompanied by the corresponding energy storage performance observed across different regions of the film at 200 °C. **f** Cyclic performance of PEI/DADQ film at 200 °C and 400 MV m⁻¹.

numerous high-temperature polymer composites that also maintain efficiency above 90%. At 200 °C, the $U_e$ value for $\eta \geq 90\%$ in other polymers are as follows: 1.8 J cm⁻³ of Al₂O₃-BT@SiO₂/PI-Al₂O₃[44], 1.8 J cm⁻³ of PEIs/PI/PEIs[45], 2.2 J cm⁻³ of o-POFNB[32], 2.6 J cm⁻³ of PEI/NTCDA[33], 3.45 J cm⁻³ of PI-3.0[14], 4.39 J cm⁻³ of F-PI/PCBM[35], 4.8 J cm⁻³ of FPE/ICIT-Cl[19], and 5.74 J cm⁻³ of AI-2 PI[18]. These values are lower than the results obtained for PEI/DADQ. This comparison highlights the significant advancement represented by the PEI/DADQ composite in high-temperature energy storage materials, positioning it as a promising candidate for applications under harsh conditions.

Large-area all-organic composite films doped with high-polarity organic molecules were fabricated using the industrial solution-casting method (Fig. 4e). Compared to alternative techniques, this method is more cost-effective and scalable, making it commercially viable for mass production. The process offers precise control over film thickness and dopant distribution, essential for ensuring uniform dielectric and mechanical properties throughout the film. The PEI/DADQ film, notable for its flexibility and transparency, is designed to assess the uniformity of its energy storage capabilities. Samples taken from different regions of the large film consistently exhibited stable energy storage performance at 200 °C and 400 MV m⁻¹, demonstrating high homogeneity. This uniformity underscores the potential of PEI/DADQ for industrial applications, emphasizing its suitability for large-scale manufacturing. Furthermore, a cyclic charge–discharge assessment

was performed to ascertain the long-term reliability of PEI composites. As illustrated in Fig. 4f, PEI/DADQ exhibited consistent capacitive performance after extensive charge–discharge cycling for up to 100,000 cycles at 200 °C and 400 MV m⁻¹. Over the entire cycle range, no degradation in $U_e$ and $\eta$ was observed for the PEI/DADQ composite, demonstrating its robustness during rapid discharge experiments under these conditions. In comparison, other composites such as PEI/DMABN, PEI/PTCNQ, and pristine PEI exhibited significantly lower cycling stability, with failures occurring after 25,119 cycles for PEI/DMABN, 15,849 cycles for PEI/PTCNQ, and just 10,001 cycles for pristine PEI. The ability to maintain high $U_e$ and $\eta$ over a large number of cycles indicates that the incorporation of DADQ into the PEI matrix effectively enhances the polymer's durability. Furthermore, a large-area PEI/DADQ film, exhibiting notable flexibility and transparency, was fabricated to evaluate the uniformity of its energy storage capabilities. All tested samples obtained from the large film exhibited stable energy storage performance at 200 °C and 400 MV m⁻¹, demonstrating high consistency across various regions and confirming the film's homogeneity. This consistency in performance exhibits the potential of PEI/DADQ for industrial applications and highlights its suitability for large-scale manufacturing processes.

In conclusion, we present a generalizable modification method to optimize capacitive energy storage performance for high-temperature polymers. Incorporating high-polarity organic molecules, such as

DADQ, DMABN, and PTCNQ, into the PEI matrix enhances the energetic disorder of the polymer composites through dipole-dipole interactions. Experimental results, in conjunction with computational simulation, illustrate that enhanced energetic disorder contributes to inhibiting charge carrier transport, which provides fundamental insights into the characterization of charge dynamics in disordered polymer dielectrics. At 200 °C, the optimized PEI/DADQ composite exhibits outstanding capacitive energy storage properties, achieving $U_e$ of 6.45 J cm$^{-3}$ with $\eta$ of 90%. The simultaneous achievement of high energy density and consistent efficiency in the all-organic composite supports the reliable operation of film capacitors under high-temperature and other challenging conditions.

## Methods

### Materials

4,4′-Methylenedianiline (MDA, 98%) and 4,4′-(4,4′-Isopropylidenediphenoxy)bis (phthalic anhydride) (BPADA, 98%) were purchased from Tokyo Chemical Industry (TCI). The dianhydride monomer BPADA was stored in a desiccator to prevent hydrolysis, while the amino monomers MDA were kept refrigerated to prevent oxidation. 4-(N,N-Dimethylamine)benzonitrile (DMABN, 97%) were obtained from Macklin. In the two-step polymerization process of PEI composites, N-methyl-2-pyrrolidinone (NMP, 99%) from Sigma-Aldrich was employed as the solvent. 7,7,8,8-Tetracyanoquinodimethane (TCNQ, 98%) from TCI was recrystallized from acetonitrile prior to use.

### Synthesis of organic molecules

For the synthesis of PTCNQ, a solution of TCNQ (250 mg, 1.22 mmol, 98%) in acetonitrile (20 mL) was warmed and stirred, followed by the addition of pyrrolidine (69.7 mg, 0.98 mmol, 99%) in a single portion. The solution color changed from green to purple. After stirring for 4 h at 70 °C, the mixture was allowed to cool to room temperature and then stored in the refrigerator for 3 days. The resulting precipitate, PTCNQ, was filtered and washed with chilled acetonitrile, yielding fine purple crystal needles (232.4 mg, 0.93 mmol, 76.5%). For the synthesis of DADQ, ethylenediamine (26.6 mg, 0.44 mmol) was added to a pre-warmed (40 °C) solution of PTCNQ in acetonitrile (10 mL). The solution rapidly transitioned from deep green to yellow. After 4 h of stirring at 70 °C, the mixture was allowed to cool to room temperature. The resulting precipitate, DADQ, was filtered and washed with chilled acetonitrile, yielding a fine-grain yellow powder(73.4 mg, 0.35 mmol, 78.8%).

### Films preparation

The composite films were prepared using the solution casting technique. Initially, a specified mass of BPADA was dissolved in 10 mL of NMP solution. An equimolar amount of MDA was added to the solution. The mixture was then stirred at room temperature for 12 h to form a poly(amic acid) (PAA) solution, with the total monomer mass consistently maintained within the range of 300–400 mg. Simultaneously, high-polarity organic molecules, such as DADQ, DMABN, and PTCNQ, were dissolved in NMP at a concentration of 5 mg mL$^{-1}$ and sonicated for 3 h. Afterwards, the NMP solution containing the high-polarity organic molecules was combined with the PAA solution in the desired proportions, resulting in the formation of the composite solution. The composite solution was then cast onto a pre-cleaned glass plate and dried at 80 °C for 12 h to remove the solvent. This was followed by sequential heating at 150 °C for 1 h, 200 °C for 1 h, and 250 °C for 1 h. Finally, after cooling naturally to room temperature, the film was peeled off from the glass substrate in deionized water and vacuum-dried in an oven at 100 °C to eliminate any residual moisture. The resultant films had a thickness maintained within the range of 9–12 μm.

### Materials characterization

ATR-FTIR spectra were obtained with a Nicolet iS10 spectrometer (Thermo Scientific) at a resolution of 4 cm$^{-1}$ [50]. UV-vis absorption measurements of the samples were conducted using a Hitachi U-3010 spectrophotometer. Thermal behavior was analyzed via differential scanning calorimetry (DSC) on a TA Instruments Q8000 system, employing 5 mg samples subjected to three consecutive heating and cooling cycles from 30 °C to 300 °C at 10 °C min$^{-1}$ under a nitrogen atmosphere. Dynamic mechanical analysis (DMA) was performed using a TA Instruments DMA Q800, with measurements conducted under a temperature ramp of 5 °C min$^{-1}$. For the mechanical property test, the samples were fabricated as elongated strips with a width of 10 mm and a length of 40 mm, maintaining a consistent effective gauge length of 30 mm across all specimens. $^1$H Nuclear Magnetic Resonance (NMR) spectra were obtained on a JNM-ECA600 spectrometer, utilizing deuterated acetone as the solvent. Small-angle X-ray scattering (SAXS) measurements were conducted at beamline 1W2A of the Beijing Synchrotron Radiation Facility (BSRF). Nano-IR measurements were conducted using a Bruker Anasys nanoIR3 system, equipped with a quantum cascade laser (QCL) source operating in the 1800–2700 cm$^{-1}$ range, in tapping AFM-IR mode. The topography and surface potential of composite film samples were characterized using a Bruker Dimension Icon scanning probe microscope.

Dielectric spectra were acquired between $10^2$ and $10^6$ Hz using a Novocontrol Concept 80 spectrometer with a temperature-controlled Quatro Cryosystem. Leakage currents were recorded at electric fields from 25 to 300 MV m$^{-1}$ using a Keithley 6517B picoammeter coupled to a voltage source. Thermally stimulated depolarization currents (TSDC) were measured with a Keithley 6517B electrometer under controlled thermal/electrical protocols. Samples were polarized under a DC electric field of 50 MV m$^{-1}$ at 200 °C for 1 h, followed by field-maintained cooling to 30 °C. After stabilization for 20 min, the electric field was removed and the samples were short-circuited. Depolarization currents were subsequently recorded during heating to 250 °C at a rate of 2 °C min$^{-1}$. D-E hysteresis loops were measured using a K-CPE1901-AI-30kV high-voltage system. Samples, immersed in dimethylsilicone fluid, were driven by a 100 Hz unipolar triangular waveform. Dielectric breakdown strength was evaluated using a TREK 610 C amplifier by applying a constant DC voltage ramp of 500 V s$^{-1}$ until electrical failure. The resulting data were analyzed via the two-parameter Weibull statistical model. Each sample was subjected to a minimum of 20 tests, with five parallel specimens tested per condition. Aluminum electrodes (3 mm diameter, 20 nm thickness) were deposited on both faces of the polymer films to enable measurements of dielectric spectra, conduction currents, D-E hysteresis loops, and breakdown strength.

### Simulation

The simulations, encompassing electrostatic potential, dipole moments of high-polarity organic molecules, and dipole-dipole interactions, were performed utilizing the GAUSSIAN software suite. This study investigated the impact of incorporating high-polarity organic molecules on the broadening of electronic energy level distributions within the polymer matrix, employing Density Functional Theory (DFT) calculations with the CAM-B3LYP functional and the aug-cc-pVTZ basis set. To facilitate a comprehensive comparison, electronic energy level distributions were calculated using the Vienna Ab Initio Simulation Package (VASP), which employs a periodic framework to model the polymer composite in three dimensions. This approach effectively captures long-range interactions, which are essential for accurately describing the bulk electronic properties of the material.

Molecular dynamics (MD) simulations were conducted using Orca 5.0.3 to investigate the effects of high-polarity small molecule doping on energy fluctuations and energy distributions. The simulations employed a timestep of 10 ps over a total duration of 2000 ps, performed at the B3LYP/def2-SVP level of theory with an initial temperature of 350 K. Moreover, we utilized the Vienna Ab Initio Simulation

Package (VASP) to conduct molecular dynamics (MD) simulations on a polymer composite system. These simulations were performed under periodic boundary conditions, enabling the modeling of the system in three dimensions and the capture of long-range interactions, such as chain cross-linking and the effects of polymer chain packing on electronic properties. The free volume of the doped polymer was also quantified using techniques such as Voronoi tessellation, illustrating how doping modifies the unoccupied spaces within the polymer matrix.

The electron decay simulation involved injecting charges into the center of the polymer with varying levels of energetic disorder and tracking their dissipation within the polymer matrix. By modulating the degree of disorder, the model investigated how charge carriers propagate through the material. To quantify the impact of energetic disorder on charge transport, three fundamental equations governing the relationship between disorder and charge mobility were solved simultaneously using COMSOL Multiphysics.

We performed the phase-field simulations incorporating the electrical-thermal-mechanical breakdown mechanism to simulate the phase breakdown evolution of electric field distributions in pristine PEI and its corresponding composites, including PEI/DADQ, PEI/DMABN, and PEI/PTCNQ. The phase-field simulations were conducted under an applied electric field of 600 MV m$^{-1}$ at 200 °C.

## Data availability

The authors declare that all the relevant data are available within the paper and its Supplementary Information file or from the corresponding author on request.

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

## Acknowledgements

This research was funded by the National Natural Science Foundation of China (grant no. 52377023 (C.Y.), 52007094 (C.Y.), 92166203 (Q.L.), and 51921005 (J.L.H.) and the Huxiang Youth Talents Plan (grant no. 2023RC3104 (C.Y.))).

## Author contributions

C.Y. and T.Z. conceived the idea. T.Z. and L.M. performed the majority of the experiments. Qiao L., and Q.Z. assisted in the sample preparation and performance tests. D.D.L. and Q.Z. performed the simulations. C.Y., Qiao L., Qi L., J.L.H., L.M., and T.Z. analyzed the data. T.Z., C.Y., and Qi L. wrote the manuscript. C.Y., Qi L., and J.L.H. supervised the whole project. All authors discussed the results and commented on the manuscript.

## Competing interests

The authors declare no competing interests.
