## [Transparent Peer Review file · Nature Communications]

Enhancing Energetic Disorder in All-organic Composite Dielectrics for High-temperature Capacitive Energy Storage

Corresponding Author: Dr Chao Yuan

Version 0:

Reviewer comments:

Reviewer #1

(Remarks to the Author)

The manuscript demonstrates increased energetic disorder arising from dipole–dipole interactions between high-dipole-moment molecules and polymer chains, as corroborated by PL spectroscopy and KPFM analyses. Further insights from TRPL mapping and ISPD measurements reveal that this enhanced energetic disorder significantly reduces carrier mobility under applied electric fields, effectively suppressing charge transport. These findings are both compelling and impactful, highlighting notable performance enhancements. The reviewer recommends minor revisions to further improve the manuscript. The detailed remarks are as follows.

1) The manuscript presents a comprehensive examination of breakdown strength and energy storage performance across a range of doping concentrations, with optimal capacitive performance observed at 0.5 vol%. However, the mechanisms through which highly polar organic molecules at different doping levels modulate the energetic disorder of polymers and subsequently influence critical parameters, including breakdown strength and energy storage properties, require further elucidation.

2) The manuscript's current density analysis is restricted to experimental data collected at 200 °C, with the applicability of its conclusions at lower temperatures, such as 150 °C, remaining unaddressed. Additionally, the lack of an investigation into the activation energy for charge transport across a broader temperature range represents a notable limitation. Including such an analysis would provide critical insights into the thermal dependence of the proposed mechanisms, significantly enhancing the study's scientific rigor and broader relevance.

3) While the manuscript provides unipolar D-E loop data, the behavior under bipolar conditions is equally critical, particularly for applications in alternating current (AC) fields. Bipolar loops are generally broader due to charge injection and recombination during electric field reversal. Including bipolar loop data would enable a more comprehensive evaluation of the material's performance under diverse operational scenarios.

4) The DSC data provided in the supplementary materials (Figure S4) appear insufficient for precise T_g estimation due to the presence of noticeable noise. Employing a higher heating rate, such as 20°C/min or 30°C/min, may mitigate the noise and improve the accuracy of the T_g determination, thereby strengthening the reliability of the reported results.

5) The characterization section requires further refinement to provide comprehensive details on the techniques employed. Specifically, the dimensions of the samples used for Young's modulus determination should be explicitly stated to ensure reproducibility. For fs-TAS and TRPL mapping, additional information on the sample preparation process is necessary, including specific measures taken to ensure surface uniformity and optical quality, as these factors could significantly influence the results.

Reviewer #2

(Remarks to the Author)

Referee Report -- NCOMMS-24-75318-T / Chao Yuan et al

The authors have presented experimental and computational studies on all-organic composite dielectric material PEI/DADQ where high-polarity organic molecules have been incorporated into the polyetherimide (PEI) matrix.

Interestingly, the authors argue that by enhancing the energetic disorder by electric dipole-dipole interactions in this composite material will suppress charge carrier transport and leads to essentially higher energy densities.

Furthermore, owing to the huge number of charge-discharge cycles (100,000 at 200 C) under very large electric field strengths order of 400 MV m^{-1} it is obvious that the PEI/DADQ composite may well become a very promising new material in high-temperature capacitive energy storage systems used in high-voltage power energy systems and electronic devices.

A definite and quite a unique merit of this manuscript stems from the fact that the authors have carried out their studies on novel all-organic polymer composites for high energy density capacitor related applications with high dielectric breakdown strength by using a wide variety of state-of-art experimental techniques (e.g. FTIR, PL and fs-TAS) and computational methods (ab initio molecular dynamics, DFT and simulation tools).

Furthermore, considering the presently ongoing search for high-quality dielectric materials suitable for high-energy storage capacitors (supercapacitors) and the promising results reported by the authors this manuscript possibly deserves to become published in Nature Communications.

Finally, from the linguistic viewpoint the manuscript has been written clearly with good-quality English.

However, there are several major serious scientific shortcomings and technical points, both in experimental and especially in theoretical settings in this manuscript.

For example, it is not quite clear how the different theoretical and experimental tools used in this study are relevant for the central topic of this study ("enhancing energetic disorder") and support or complement each others. Furthermore, no experimental characterization has been provided in order to quantify the 3D crystallographic nature of the PEI + organic high-polarity molecules -polymer composites.

Providing that the authors manage to address these scientific shortcomings and technical points, and consequently carry out further experiments and calculations it might be possible to accept the revised manuscript to become published in Nature Communications.

1. Section "Introduction":

"Energetic disorder" in pristine polyetherimide (PEI) polymer and in PEI alloyed or doped with high-polarity organic molecules is one of the most central concepts discussed or referred throughout in this manuscripts.

However, the authors have given no accurate definition and description of its microscopic origin (for example, how it is related to the STRUCTURAL DISORDER in the polymer composite) in their manuscript.

Instead, the authors try to describe the meaning of "energetic disorder" by some quite indirect and unclear expressions, such as:

"Energetic disorder refers to the variations and fluctuations in energy states within the polymer structure, typically characterized by a Gaussian distribution of the density of states (DOS)." (page 3)

This sentence prompts the following questions and revision tasks:

(a) The authors should clarify the concept of "energetic disorder" and briefly describe the nature of "energy states within the polymer structure" in their revised manuscript.

Do the authors refer in this context, for example, to electronic (quasi-particle) energy states or some local polymer structural (macro)molecular-like energy states, meaning a total energy of some local part of the polymer chain including randomly distributed high-polarity organic molecules (dopants)?

(b) Furthermore, it does not become clear how the "energetic disorder" is distributed in the polymer structure, whether it's occurring only along the 2-dimensional surface of the polymer film, or also distributed vertical to it. The authors should clarify also this point.

(c) Finally, the (continuous) DOS concept is only valid for infinite materials systems (periodic, amorphous, polycrystalline etc),

Therefore, when the authors are representing the PEI + organic high-polarity molecules -systems using molecular modelling (e.g. molecular ab initio GAUSSIAN software package) it is fundamentally NOT correct to use the concept of DOS. The authors are advised to use some other terminology or explain with a notice the use of "DOS"

in their molecular studies context.

2. Section "Results and discussions / Modulation of energetic disorder":

The authors write "Energetic disorder arises from the variations in the local energy landscape, influenced by intermolecular forces such as van der Waals interactions, hydrogen bonds, and dipole-dipole interactions". (on page 3)

As requested before the authors should clarify where does the expression "local energy landscape" refer to? Does it refer to some locally estimated single particle (electronic, quasi-particle) energies or total energies of local portions (say, within fixed-sized computational unit cells) of the polymer structure randomly doped with high-polarity molecules?

Concerning the microscopical origin of "energetic disorder" why the authors have not mentioned intramolecular forces which they include while mentioning "significant intramolecular charge transfer between the high-polarity molecules and the PEI chains"? (on page 4)

3. Section "Results and discussions / Modulation of energetic disorder":

From the sentence

"... indicate a significant intramolecular charge transfer between the high-polarity molecules and the PEI chains, ..." (on page 4)

it does not become clear how the high-polarity molecules are topologically distributed (randomly) into the PEI polymer structure.

For example, have they been grafted along the polymer chains or located in between the chains?

Furthermore, should the word "intramolecular" be replaced by the word "intermolecular"?

The authors should clarify these points in their revised manuscript.

4. Section "Results and discussions" / "Characterization of charge carrier dynamics":

The authors write on page 5:

(a) "As shown in Fig. 2a, we compared the wide-field PL intensity images of PEI/DADQ, PEI/DMABN, PEI/PTCNQ, and pristine PEI, recorded at 390 nm."

(b) "Spatial energetic disorder in the films is evidenced by variations in the PL spectrum resulting in different spatial distributions of PL intensity at the same wavelength."

(c) "Whereas the PL intensity images of pristine PEI display uniform spatial distributions at 390 nm, the PEI/DADQ, PEI/DMABN, and PEI/PTCNQ composites show distinct and varied spatial distributions at this wavelength."

These three sentences raise the following questions, respectively, which the authors should comment also in their revised manuscript:

Q(a) The authors should mention, preferably in the figure caption, what is the PL excitation wavelength?

Is 390 nm the PL emission wavelength?

Why the authors have chosen these particular PL excitation and emission wavelengths?

Q(b) From the electronic structural variations' viewpoint (which probably best indicate "energetic disorder") one would get a better fingerprint

if one measures several PL spectra as a function of the photon emission energy across the polymer thin film surface rather than

measuring only one PL intensity value (or the averaged PL intensity value over emission energies) per one spatial point on the film.

Therefore, instead of measuring PL intensity variations on the polymer films could the authors measure PL spectral variations on the

polymer films and interpret PL spectral features (electronic / excitonic transitions) as well?

This would enhance the scientific level of the present manuscript tremendously.

Q(c) Could authors explain what molecular structural / electronic features possibly cause variations in spatial distributions of

the
PL intensity (spectra) in the PEI/DADQ, PEI/DMABN and PEI/PTCNQ thin film composites?

5. Section "Results and discussions" / "Characterization of charge carrier dynamics":

The authors write on page 5:

"To quantify the underlying energetic disorder within the composites, we computed the PL intensity and generated corresponding histograms from the spatial maps, as shown in Fig. 2b. "

There are a few unclear points in this sentence which the authors should clarify in their revised manuscript:

(a) Is the "PL intensity" computed for a fixed emission wavelength (what is that wavelength) or has it been integrated over all PL emission wavelengths?

(b) What is the underlying theory and computer code to calculate the PL intensity?

(c) For each PEI + organic high-polarity molecules system (thin polymer composite film) does the broadening parameter σ of the "PL intensity" have the same value for all directions along the polymer thin film surface? What would this indicate about the distribution of polymer chains and/or organic dopant molecules inside the thin polymer film?

6. Section "Methods" / "Materials characterization ":

The authors have utilized several state-of-the-art experimental facilities to characterize local (molecular) electronic and vibrational features of the PEI + high-polarity organic molecules (DADQ, DMABN, PTCNQ) -polymer composites.

However, no experimental characterization methods have been used in order to expose long-range ordered or disordered structural features of these polymer composites (e.g. crystallization: lamellae and spherulites) and cross-linking between the chains chains. This is a serious scientific shortcoming of this manuscript because it is obvious that these long-range structures have an important impact on electronic, dielectric and charge transport properties studied in this manuscript.

An ideal tool to probe long-range structures of polymers would be an X-ray synchrotron radiation facility where both the photon wavelength and polarization state can be tuned or in the simplest approach using just a standard fixed-wavelength X-ray diffractometer.

The authors are requested to comment in their revised manuscript this important point about their missing experiments on long-range structural features in their polymer composites.

Related to this point, the authors are also requested to carry out experimental X-ray diffraction studies on their polymer composite samples by using either a fixed-wavelength diffractometer or preferably using a resonant (anomalous) X-ray scattering technique with a synchrotron radiation facility which would reveal atom-specific features inside the the long-range 3D structure of the polymer composites. X-ray radiation facilities are available, for example, in France (Grenoble, ESRF) and Sweden (Lund, MAX IV). These extra experiments would greatly enhance the scientific level of this manuscript.

7. Section "Methods" / "Simulation":

By using Density Functional Theory (DFT) based molecular modelling (GAUSSIAN software package) the authors calculate electric dipole moments of high-polarity organic molecules, their dipole-dipole interactions and broadening of the electronic density of states (DOS) in the polymer structure where these high-polarity molecules have been incorporated into the PEI polymer matrix.

There are two fundamental shortcomings which make this theoretical approach, in principle, incorrect.

First of all, in order to tackle the electronic properties of the 3-dimensional (3D) PEI + high-polarity organic molecules - polymer systems correctly one should exploit a DFT computational method suitable for 3D amorphous or periodic polymer systems where, for example, cross-linking effects onto electronic structure between polymer chains can be included. Among such first-principles DFT and beyond -codes which is capable to tackle electronic, magnetic, optical

and phonon properties of a wide range of metals, insulators and semiconductors is the VASP package. I wonder if the authors could complement their present computational studies by carrying out some of their calculations on the 3-dimensional (3D) PEI + high-polarity organic molecules -polymer composites by using VASP or some other relevant DFT code (CASTEP, ABINIT, QUANTUM ESPRESSO etc) for 3D materials systems? It would be also interesting to know how the results then change from those computed using molecular modelling (GAUSSIAN).

Secondly, concerning the broadening of the electronic DOS due to randomly doping or alloying PEI polymer matrix with high-polarity organic molecules one should fundamentally employ some quantum mean-field theory model, for example, the coherent potential approximation (CPA) or average-t-matrix approximation (ATA) which involve approximations to average the disordered electronic Green's function or self-energy.

From the present manuscript it looks like the authors have ignored using any mean-field approach, even for the simplest case: organic high-polarity molecules distributed randomly along an isolated PEI polymer chain? However, the lack of mean-field theory can be approximately replaced by choosing large enough computational unit cell = supercell (3D periodic DFT modelling) or when using GAUSSIAN randomly placing organic high-polarity molecules along a sufficiently long piece of PEI polymer chain.

Could the authors carry out some extra DFT calculations along these lines (using either 3D periodic code or GAUSSIAN molecular code) and then report in their revised manuscript the randomness broadening (and perhaps electronic energy level shifting) effect(s) on the DOS? This would greatly enhance the scientific level of this manuscript.

8. Section "Methods" / "Simulation":

The authors carried out *ab initio* molecular dynamics (MD) simulations by using Orca software package (version 5.0.3). However, the MD module of Orca is meant for molecular dynamics studies on finite, non-periodic molecular systems only rigorously speaking is not valid for infinite amorphous or periodic 3D systems with long-range electronic interactions as is case with the PEI/PTCNQ, PEI/DMABN and PEI/DADQ polymer composites.

For example, how can the cross-linking interaction between the polymer composite chains be included into the MD simulation in Orca? In contrast, the VASP code contains the possibility to run *ab initio* MD for infinite and periodic 3D materials systems by utilizing the supercell method.

Could the authors run their MD simulations by using also the VASP code and then compare the results to those carried out by using Orca?

The authors are requested to comment this point related to their MD simulations and report their possibly extra MD calculations carried out by using the VASP code in their revised manuscript.

9. Section "Methods" / "Simulation":

The authors write

"By modulating the degree of disorder, the model investigated how charge carriers propagate through the material. To quantify the impact of energetic disorder on charge transport, three fundamental equations governing the relationship between disorder and charge mobility were solved simultaneously using COMSOL Multiphysics."

In this context the authors should briefly explain in the revised manuscript or give a reference to how the "modulation of the degree of disorder" theoretically has been implemented and in what directions of the polymer composite film the charge transport occurs (along the surface / vertical to the surface / along certain crystallographic axes etc).

Furthermore, the authors should explicitly describe the "three fundamental equations ..." mentioned above. The authors should also describe in their revised manuscript at what level of theory they have modelled the charge transport (e.g. semi-classical Boltzmann transport theory etc) in the "PEI + organic high-polarity molecules" -polymer composites by using the COMSOL Multiphysics software package.

10. Section "Methods" / "Simulation":

The authors write

"We performed the phase-field simulations incorporating electrical-thermal-mechanical breakdown mechanism to simulate the phase breakdown evolution of electric field distributions in pristine PEI and its corresponding composites, including PEI/DADQ, PEI/DMABN, and PEI/PTCNQ."

This sentence has been written at a very general level and therefore leads to a very unclear picture how the breakdown evolution in the polymer composite films has been calculated, raising the following questions:

(a) The authors should give some relevant references concerning the theory they use for the phase-field simulations and also mention at what level this theory has been set up (what classical, semi-classical features or even quantum microscopical features have been included?).

The harsh reality is that to the large extent the full quantum dynamical (non-equilibrium) theory on dielectric breakdown phenomenon for insulator materials has still not been developed.

(b) The authors should briefly describe the main underlying mathematical features of the model they use to approximate their electrical breakdown phase calculation in Fig. 3h:

I assume $\eta(r,t)$ refers to "phase-field" (auxiliary field in the theory of phase-field model)?

How this auxiliary field has been constructed mathematically from the "electrical", "thermal" and "strain" induced breakdown physics concepts?

The functional variables f_{sep} , f_{grad} , f_{elec} , f_{Joule} and f_{strain} and the constant L_0 of Eq. (S3) [see authors' Supplementary file] should be explained in more detail.

How the additive form of $f_{\text{sep}} + f_{\text{grad}} + \dots$ in Eq. (S3) can be justified in the breakdown situation of PEI composite polymers?

(c) It is also quite confusing that in the manuscript text on page 13 and Supplementary file's text in connection with Eq. (S3) the authors are talking about "electrical-thermal-mechanical" breakdown phase while in caption to Fig. 3h they refer only to "electrical breakdown phase". The authors should describe in detail how the "electrical" breakdown phase model can be extracted from the "unified" "electrical-thermal-mechanical" breakdown model of Eq. (S3).

Reviewer #3

(Remarks to the Author)

This paper reveals that the incorporation of high-polarity organic molecules into a polyetherimide (PEI) matrix, demonstrating that dipole-dipole interactions enhance the energetic disorder of the material. By integrating experimental observations with computational simulations, the authors establish that the increased energetic disorder significantly suppresses charge carrier transport, offering valuable insights into the charge dynamics of disordered polymer dielectrics. This study presents innovative and significant findings, rendering it highly pertinent and engaging for researchers in the field of dielectric materials. I recommend to accept the paper for publication after the following issues are addressed.

i. Molecular weight (Mw) is a key determinant of the electrical and mechanical properties of polymers, yet its potential influence is not explored in this study. The authors are strongly encouraged to evaluate and compare the Mw of pristine PEI, PEI/DADQ, PEI/DMABN, and PEI/PTCNQ composites to assess whether Mw plays a significant role in the observed properties or can be reasonably excluded as a factor in this context.

ii. Film capacitors utilize the inherent versatility of polymers, enabling their transformation into thin films that can be rolled into compact configurations, emphasizing the crucial relationship between polymer mechanical properties and device performance. The Supporting Information includes measurements of Young's modulus, which are not addressed in the main text. The authors are encouraged to clarify the purpose of these measurements and their relevance to the study. Additionally, a more comprehensive discussion on the polymer's softness or flexibility and its influence on mechanical behavior and capacitor performance would provide valuable insights and strengthen the manuscript.

iii. The synthesis of polyimide (PI) involves a complex imidization reaction, with the thermal imidization process playing a pivotal role in determining the dielectric and energy storage properties of the material. The authors are encouraged to provide clarity on the degree of imidization achieved at 250 °C and confirm whether the imidization process is complete under these conditions.

iv. The manuscript describes the incorporation of high-polarity organic molecules into a viscous PAA solution but lacks details regarding the dispersion process and the stability of the organic molecules within the solution. The authors are encouraged to provide a comprehensive description of the methodology used for introducing the organic molecules, evaluate their dispersibility, and address any potential challenges encountered during this step.

v. Please provide a comprehensive description of the phase-field simulation methodology employed for analyzing the evolution of the electrical breakdown phase in this study.

vi. The characterization section would benefit from additional refinement to offer a more comprehensive description of the tested samples, thereby enhancing clarity and ensuring reproducibility. In particular, detailed information on the Nano-IR and KPFM measurements should be provided, with an emphasis on the sample preparation protocols employed.

Version 1:

Reviewer comments:

Reviewer #1

(Remarks to the Author)

The comments have been well addressed. It can be accepted.

Reviewer #2

(Remarks to the Author)

Referee Report -- NCOMMS-24-75318A / Tan Zeng et al

I (as Referee 2) have carefully viewed the authors' response letter, their revised manuscript as well as the comments and recommended changes of the other two referees (Referee 1 and 3).

In their response (rebuttal) letter the authors have commented diligently in very detail and extensively to my (Referee 2) several questions and requests to modify the manuscript.

Furthermore, in my opinion, the authors have also responded very well and in detail to the questions and requests of Referees 1 and 3 to improve the manuscript including extra experimental analysis and calculations and also incorporated the relevant modifications to the manuscript.

However, several discussion points about DFT electronic structure and molecular dynamics (MD) related extra calculations to model 3D (periodic / disordered) polymer composites that the authors describe extremely well in their response letter are **TOTALLY MISSED OUT** in their revised manuscript.

The authors need to be address these points (see below) in their second revised manuscript before it can be published in the journal of Nature Communications.

Finally, as prompted by the Associate Editor I have composed a minor question related to the usage of the Nano-IR experiments (see below).

1. Authors' response letter (559842_1_rebuttal_10311778_srtfld.pdf):

On page 31 the authors write spot on correctly:

"Given the localized nature of these interactions, our GAUSSIAN-based modeling approach provides an effective framework for capturing the key electronic modifications at the molecular level. Nonetheless, we recognize the importance of extending our analysis to encompass long-range effects within the polymer matrix, which can be more comprehensively addressed through periodic DFT calculations. This perspective has been acknowledged in the revised manuscript, and future efforts will focus on bridging the molecular and periodic modeling approaches to gain a more holistic understanding of the composite system."

However, the authors seem NOT to mention anything about this in their revised manuscript (559842_1_art_file_10323768_srbvtg.pdf).

The authors are advised to give these important comments in their revised manuscript, preferably within the Section "Simulation".

2. Authors' response letter (559842_1_rebuttal_10311778_srtfhd.pdf):

On page 32 the authors have responded extremely well and spot on to my questions by stating correctly:

"While a full implementation of CPA/ATA methodologies may extend beyond the scope of this study, we have adopted a series of alternative computational strategies to approximate disorder-induced effects within the polymer composite system. To gain meaningful insights into the impact of energetic disorder and the intricate polymer-dopant interactions, we employed supercell modeling within the VASP framework. This approach allows for the simulation of the stochastic distribution of dopants and the corresponding broadening of electronic states within a three-dimensional environment. Complementarily, GAUSSIAN-based molecular simulations were conducted with randomly distributed high-polarity dopants along extended PEI polymer chains, enabling the investigation of localized electronic perturbations and charge distribution effects. Additionally, a comparative analysis of the electronic density of states (DOS) obtained from both GAUSSIAN molecular simulations and 3D periodic DFT calculations provides valuable insights into the role of structural disorder and facilitates the cross-validation of computational findings."

and on page 33 the authors continue to write equally importantly:

"In the revised manuscript, we have expanded the discussion to outline the theoretical rationale, limitations, and practical relevance of these methodologies in approximating mean-field effects, thereby enhancing the scientific depth of our study and addressing the reviewer's concerns."

and

"In response, we have expanded the discussion in the revised manuscript to provide a comprehensive comparison between the molecular modeling results obtained using GAUSSIAN and those derived from periodic DFT calculations."

However, NONE of these important texts or DFT calculations (VASP, GAUSSIAN) exist in the revised manuscript (559842_1_art_file_10323768_srbvtg.pdf).

The authors are requested to show these theoretical electronic structure methodologies related comments also in their revised manuscript, preferably within the Section "Simulation".

3. Authors' response letter (559842_1_rebuttal_10311778_srtfhd.pdf):

On page 34 the authors have responded extremely well and spot on to my questions by stating correctly:

"The ORCA MD module was employed in this study to gain molecular-level insights into the local interactions between polymer chains and high-polarity dopant molecules. This approach enables a detailed examination of localized phenomena, such as molecular conformations, dipole-dipole interactions, and short-range structural fluctuations within the polymer matrix. However, we acknowledge that the absence of periodic boundary conditions in ORCA-based MD simulations limits their ability to capture long-range structural features, such as chain cross-linking and spatial heterogeneity—factors that are crucial for accurately characterizing the dielectric and charge transport properties of the polymer composites. To address these limitations, we have revised the manuscript to provide a clearer rationale for the selection of ORCA, emphasizing its suitability for probing localized molecular-scale interactions rather than bulk material properties."

However, NOTHING along these lines have been discussed in the revised manuscript (559842_1_art_file_10323768_srbvtg.pdf).

The authors are requested to include this important methodological discussion also in their revised manuscript, preferably within the Section "Simulation".

4. Authors' response letter (559842_1_rebuttal_10311778_srtlfd.pdf):

On pages 34-35 the authors have responded extremely well and spot on to my questions by stating correctly:

"In response to the reviewer's insightful suggestion, we have expanded our computational analysis by conducting additional molecular dynamics (MD) simulations using the Vienna Ab initio Simulation Package (VASP). The implementation of periodic boundary conditions and the supercell approach within VASP enables a more accurate and holistic representation of the polymer composite system, capturing long-range electronic interactions and structural dynamics. These simulations provide deeper insights into the effects of polymer chain cross-linking on charge distribution and structural integrity, offering a more realistic depiction of the amorphous polymer matrix, including density fluctuations and packing effects. The revised manuscript now includes a detailed discussion of the VASP-based MD simulations, outlining the system configuration, computational parameters, and a comparative evaluation with the previously employed ORCA-based simulations."

However, NOTHING along these lines have been discussed in the revised manuscript. Furthermore, NO VASP MD calculations the authors state to have carried out seem to exist in the revised manuscript (559842_1_art_file_10323768_srbvtg.pdf).

The authors are requested to include this important methodological discussion also in their revised manuscript, preferably within the Section "Simulation". Furthermore, the authors are requested to display, at least some of their VASP MD computed results and compare these on their ORCA molecular modeling based MD simulations in their revised manuscript.

5. Authors' response letter (559842_1_rebuttal_10311778_srtlfd.pdf):

On page 35 the authors write:

"A comparative analysis of the simulation results obtained from ORCA and VASP has been incorporated into the revised manuscript to assess their respective contributions and ensure the robustness of the findings."

However, NOTHING along these computational results have been discussed or shown in the revised manuscript. (559842_1_art_file_10323768_srbvtg.pdf).

The authors are requested to discuss, even briefly, these ORCA and VASP MD simulation results in their revised manuscript.

6. Section "Results and discussions / Modulation of energetic disorder":

On page 5 of the revised manuscript (559842_1_art_file_10323768_srbvtg.pdf) the authors write:

"As demonstrated in the Nano-IR images of the polymers (Fig.1g), the notable difference is the detection of absorption at 2140 cm^{-1} ($\text{C}\equiv\text{N}$ stretch) at the PEI/DADQ compared to pristine PEI, indicating that the DADQ molecule exhibits uniform dispersion within the polymer matrix."

In viewing the possible electronic structural and molecular conformational surface sensitivities on the PEI/DADQ polymer composite film surface I wonder how surface sensitive the Nano-IR spectroscopy will be? In other words, how surface sensitive the vibrational excitation frequencies of the DADQ molecule, such as $\nu = 2140 \text{ cm}^{-1}$, could be and why in the context of the PEI/DADQ polymer film?

The authors are advised to comment on this point also in their revised manuscript.

Reviewer #3

(Remarks to the Author)

The manuscript has been well revised. I recommend it to be accepted.

REVIEWER COMMENTS

Manuscript ID: NCOMMS-24-75318-T

TITLE: Enhancing Energetic Disorder in All-organic Composite Dielectrics for High-temperature Capacitive Energy Storage

Reviewer #1 (Remarks to the Author):

The manuscript demonstrates increased energetic disorder arising from dipole–dipole interactions between high-dipole-moment molecules and polymer chains, as corroborated by PL spectroscopy and KPFM analyses. Further insights from TRPL mapping and ISPD measurements reveal that this enhanced energetic disorder significantly reduces carrier mobility under applied electric fields, effectively suppressing charge transport. These findings are both compelling and impactful, highlighting notable performance enhancements. The reviewer recommends minor revisions to further improve the manuscript. The detailed remarks are as follows.

1) The manuscript presents a comprehensive examination of breakdown strength and energy storage performance across a range of doping concentrations, with optimal capacitive performance observed at 0.5 vol%. However, the mechanisms through which highly polar organic molecules at different doping levels modulate the energetic disorder of polymers and subsequently influence critical parameters, including breakdown strength and energy storage properties, require further elucidation.

Response: We sincerely thank the reviewer for highlighting the need for further clarification regarding the mechanisms through which varying doping levels of highly polar organic molecules influence the energetic disorder of polymers and, in turn, affect critical parameters such as breakdown strength and energy storage performance. We recognize that a more comprehensive understanding of these mechanisms is essential to the depth and overall impact of our study.

Doping a polymer matrix with highly polar organic molecules, such as DADQ, DMABN, and PTCNQ, induces dipole-dipole interactions between the dopants and the polymer chains. These interactions disrupt the local energy landscape of the polymer, resulting in leading to an increase in energetic disorder. At lower doping concentrations, the high-polarity molecules interact with the polymer, altering the electron cloud distribution of the polymer chains and thereby enhancing the degree of energetic disorder. However, as the doping concentration increases, the higher density of polar molecules leads to overlapping interactions, which reduce fluctuations in the energy landscape and consequently diminish the extent of energetic disorder. At optimal concentrations, such as 0.5 vol%, the high-polarity molecules are uniformly distributed,

promoting local energy fluctuations that facilitate an increase in the polymer's energetic disorder. This behavior is corroborated by the observed variations in breakdown field strength and discharge energy density as a function of doping concentration.

As illustrated in the chart below, both breakdown strength and discharged energy density, with efficiencies exceeding 90%, initially increase with increasing doping concentrations before subsequently declining. Remarkably, the peak values for breakdown strength and discharged energy density, while maintaining efficiencies above 90%, are observed at optimal doping concentrations (0.5 vol%) of 150 and 200°C, respectively. The modulation of energetic disorder at varying doping concentrations directly influences key parameters such as breakdown strength and energy storage performance. At lower doping levels, the enhancement of energetic disorder strengthens the breakdown strength by impeding electron transport within the composite, thereby improving the insulating properties of the material. However, beyond an optimal doping threshold, the reduction in energetic disorder leads to a decline in breakdown strength and energy storage performance. The intricate relationship between energetic disorder and doping concentration is fundamental in governing the material's overall performance. By precisely modulating the doping level, it is possible to optimize the energetic disorder, thereby enhancing the composite's electrical insulation properties.

We sincerely thank the reviewer for their thoughtful and constructive feedback. We believe that the revisions made in response to their comments have strengthened the manuscript, and we trust that the updated version will meet their expectations.

2) The manuscript's current density analysis is restricted to experimental data collected at 200 °C, with the applicability of its conclusions at lower temperatures, such as 150 °C, remaining unaddressed. Additionally, the lack of an investigation into the activation energy for charge transport across a broader temperature range represents a notable limitation. Including such an analysis would provide critical insights into the thermal dependence of the proposed mechanisms, significantly enhancing the study's scientific rigor and broader relevance.

Response: We sincerely thank the reviewer for their thoughtful comments regarding the thermal dependence of charge transport and the activation energy analysis. We agree that extending the current analysis to include data collected at lower temperatures, such as 150 °C, and investigating the activation energy for charge transport over a broader temperature range would provide valuable insights into the thermal behavior of the proposed mechanisms.

In response to this suggestion, we have expanded our measurements to include leakage current density data at 150 °C. We specifically analyzed the correlation between current density and the applied electric field across various composites, including pristine PEI, PEI/DADQ, PEI/DMABN, and PEI/PTCNQ. The resulting fitting curves consistently adhered to the hopping conduction model, suggesting that charge transport in these materials is primarily governed by hopping conduction at 150 °C. These additional data points offer crucial insights into the temperature-dependent behavior of the composites, thereby enhancing the applicability of our conclusions over a wider thermal range. This extension further strengthens the robustness of our findings and broadens the relevance of the proposed charge transport mechanisms.

Moreover, we recognize the critical role of investigating the activation energy for charge transport in elucidating the thermally activated nature of charge transport mechanisms. To address this, we conducted an Arrhenius analysis on the electrical conductivity data measured across a range of temperatures. By evaluating the temperature dependence of charge transport, we aim to extract the activation energy and provide a more comprehensive view of how doping concentrations modulate the charge transport properties across different thermal conditions. Activation energies associated with carrier transport were determined through an Arrhenius-type analysis of temperature-dependent electrical conductivity (σ). The relationship between electrical conductivity and temperature can be expressed by the Arrhenius equation:

$$\sigma(T) = \sigma_0 \exp\left(-\frac{E_a}{k_B T}\right)$$

Where $\sigma(T)$ is the electrical conductivity at temperature T , σ_0 is the pre-exponential factor, E_a is the activation energy, k_B is the Boltzmann constant and T is the absolute temperature. The activation energy E_a is obtained from the slope of the $\ln(\sigma)$ vs. $1/T$ plot. Notably, PEI/DADQ demonstrates the highest activation energy (0.96 eV) compared to pristine PEI (0.62 eV), PEI/PTCNQ (0.74 eV), and PEI/DMABN (0.86 eV).

We believe that this additional analysis will significantly strengthen the scientific rigor of our study by offering a more comprehensive understanding of the thermal dependence of energetic disorder and its influence on charge transport. In response to the reviewer's insightful suggestion, we have conducted the necessary additional analyses and revised the manuscript to incorporate these findings. This revision ensures a more comprehensive treatment of the complexities associated with thermal activation and its influence on the overall performance of the materials. We sincerely appreciate the reviewer's attention to this important aspect, as it has allowed us to further refine our work and enhance the depth of the study.

3) While the manuscript provides unipolar D-E loop data, the behavior under bipolar conditions is equally critical, particularly for applications in alternating current (AC) fields. Bipolar loops are generally broader due to charge injection and recombination during electric field reversal. Including bipolar loop data would enable a more comprehensive evaluation of the material's performance under diverse operational scenarios.

Response: We appreciate the reviewer for emphasizing the significance of incorporating bipolar D-E loop data, especially for applications in alternating current (AC) fields. We acknowledge that understanding the material's behavior under bipolar conditions is essential for accurately assessing its performance in practical applications, where alternating electric fields are routinely applied.

In response to this insightful suggestion, we have incorporated a comprehensive analysis of the bipolar D-E loops and energy storage performance for the PEI/DADQ composites at both 150 °C and 200 °C. These additional measurements, conducted at a frequency of 100 Hz, were aimed at assessing the impact of bipolar electric fields on the material's overall capacitive behavior. The results reveal that both the discharged energy density and efficiency of the bipolar PEI/DADQ composites remain virtually

unchanged compared to their unipolar counterparts at 150 °C and 200 °C. This behavior indicates that the material exhibits stable performance even under alternating electric fields, thereby enhancing its potential for AC field applications. These findings offer a more thorough evaluation of the material's performance across a range of operational conditions, further broadening its applicability.

We have revised the manuscript to incorporate these findings, enhancing the depth and breadth of our study by offering a more comprehensive understanding of the material's performance under various operational conditions. We are grateful to the reviewer for their insightful suggestion, which has significantly contributed to the overall rigor of our analysis.

4) The DSC data provided in the supplementary materials (Figure S4) appear insufficient for precise T_g estimation due to the presence of noticeable noise. Employing a higher heating rate, such as 20°C/min or 30°C/min, may mitigate the noise and improve the accuracy of the T_g determination, thereby strengthening the reliability of the reported results.

Response: We would like to express our sincere gratitude to the reviewer for the insightful comments on the DSC data presented in the Supplementary Information. We appreciate the suggestion to employ higher heating rates in order to reduce noise and improve the accuracy of the T_g determination. In response, we have conducted

additional DSC measurements on pure PEI, PEI/DADQ, PEI/DMABN, and PEI/PTCNQ composites at cooling rates of 20 °C/min and 30 °C/min. The results demonstrate that the glass transition temperatures (T_g) for these composites remain consistent, thereby confirming the reliability and reproducibility of the T_g values even at these elevated heating rates. These new data reinforce the robustness of our original measurements and further validate the reported T_g values. We have updated the supplementary materials to include these additional results, which we believe effectively address the reviewer's concerns regarding the noise in the initial DSC measurements. We greatly appreciate the reviewer's constructive input, which has significantly contributed to the refinement of our data and the overall enhancement of the manuscript. Thank you once again for your thoughtful and valuable suggestions.

5) The characterization section requires further refinement to provide comprehensive details on the techniques employed. Specifically, the dimensions of the samples used for Young's modulus determination should be explicitly stated to ensure reproducibility. For fs-TAS and TRPL mapping, additional information on the sample preparation process is necessary, including specific measures taken to ensure surface uniformity and optical quality, as these factors could significantly influence the results.

Response: Thank you for your thoughtful feedback regarding the characterization

section. We greatly appreciate the suggestions to further clarify and refine the details of the experimental procedures.

In response to the reviewer's comment regarding the dimensions of the samples used for Young's modulus determination, we have explicitly provided the precise sample dimensions in the revised manuscript. For the mechanical property test, the samples were fabricated as elongated strips with a width of 10 mm and a length of 40 mm, maintaining a consistent effective gauge length of 30 mm across all specimens. The mechanical testing was performed until sample failure, with continuous recording of stress and strain data throughout the process. This added information enhances the reproducibility of the measurements and offers a more comprehensive understanding of the experimental setup.

For the fs-TAS and TRPL mapping, we acknowledge the reviewer's comment regarding the need for further details on the sample preparation process for fs-TAS and TRPL mapping to ensure the reliability of the results. In response, we have provided an expanded description of the sample preparation procedures in the revised Supplementary Information.

The preparation of samples for both fs-TAS and TRPL mapping involves a series of critical steps, meticulously designed to ensure the precision and reproducibility of the measurements. Initially, a 2 cm × 2 cm quartz glass substrate was selected for its outstanding optical transparency and stability, making it an optimal base for subsequent film deposition and optical analysis. To prepare the poly(amic acid) (PAA) solution, a specified mass of BPADA and MDA was dissolved in 10 mL of NMP, and the mixture was stirred at room temperature for 12 hours. Concurrently, high-polarity organic molecules, such as DADQ, DMABN, and PTCNQ, were dissolved in NMP at a concentration of 5 mg/mL and sonicated for 3 hours. The solution containing these organic molecules was then combined with the PAA solution in the desired proportions, resulting in a composite solution. This composite solution was used to cast a thin film onto the quartz glass substrate via spin-coating. The film thickness was carefully controlled by adjusting both the solution concentration and spin-coating parameters. Specifically, the composite solution was spin-coated at 3000 rpm on a heated spin platform (set to 80 °C), yielding a film with an approximate thickness of 200 nm. Finally, the film was dried in a vacuum oven at 200 °C for 12 hours. This rigorous sample preparation protocol ensures that the films possess high optical quality and surface uniformity, essential for obtaining reliable results in both fs-TAS and TRPL mapping experiments.

We believe that these revisions will address your concerns and improve the clarity of the characterization section. Thank you again for your valuable suggestions, which have contributed to enhancing the quality of the manuscript.

Reviewer #2 (Remarks to the Author):

The authors have presented experimental and computational studies on all-organic composite dielectric material PEI/DADQ where high-polarity organic molecules have been incorporated into the polyetherimide (PEI) matrix. Interestingly, the authors argue that by enhancing the energetic disorder by electric dipole-dipole interactions in this composite material will suppress charge carrier transport and leads to essentially higher energy densities. Furthermore, owing to the huge number of charge-discharge cycles (100,000 at 200 C) under very large electric field strengths order of 400 MV m^{-1} it is obvious that the PEI/DADQ composite may well become a very promising new material in high-temperature capacitive energy storage systems used in high-voltage power energy systems and electronic devices. A definite and quite a unique merit of this manuscript stems from the fact that the authors have carried out their studies on novel all-organic polymer composites for high energy density capacitor related applications with high dielectric breakdown strength by using a wide variety of state-of-art experimental techniques (e.g. FTIR, PL and fs-TAS) and computational methods (ab initio molecular dynamics, DFT and simulation tools). Furthermore, considering the presently ongoing search for high-quality dielectric materials suitable for high-energy storage capacitors (supercapacitors) and the promising results reported by the authors this manuscript possibly deserves to become published in Nature Communications. Finally, from the linguistic viewpoint the manuscript has been written clearly with good-quality English. However, there are several major serious scientific shortcomings and technical points, both in experimental and especially in theoretical settings in this manuscript. For example, it is not quite clear how the different theoretical and experimental tools used in this study are relevant for the central topic of this study ("enhancing energetic disorder") and support or complement each other. Furthermore, no experimental characterization has been provided in order to quantify the 3D crystallographic nature of the PEI + organic high-polarity molecules -polymer composites. Providing that the authors manage to address these scientific shortcomings and technical points, and consequently carry out further experiments and calculations it might be possible to accept the revised manuscript to become published in Nature Communications.

1. Section "Introduction":

"Energetic disorder" in pristine polyetherimide (PEI) polymer and in PEI alloyed or doped with high-polarity organic molecules is one of the most central concepts discussed or referred throughout in this manuscript. However, the authors have given no accurate definition and description of its microscopic origin (for example, how it is related to the STRUCTURAL DISORDER in the polymer composite) in their

manuscript.

Instead, the authors try to describe the meaning of "energetic disorder" by some quite indirect and unclear expressions, such as: "Energetic disorder refers to the variations and fluctuations in energy states within the polymer structure, typically characterized by a Gaussian distribution of the density of states (DOS)." (page 3)

This sentence prompts the following questions and revision tasks:

(a) The authors should clarify the concept of "energetic disorder" and briefly describe the nature of "energy states within the polymer structure" in their revised manuscript. Do the authors refer in this context, for example, to electronic (quasi-particle) energy states or some local polymer structural (macro)molecular-like energy states, meaning a total energy of some local part of the polymer chain including randomly distributed high-polarity organic molecules (dopants)?

(b) Furthermore, it does not become clear how the "energetic disorder" is distributed in the polymer structure, whether it's occurring only along the 2-dimensional surface of the polymer film, or also distributed vertical to it. The authors should clarify also this point.

(c) Finally, the (continuous) DOS concept is only valid for infinite materials systems (periodic, amorphous, polycrystalline etc.), Therefore, when the authors are representing the PEI + organic high-polarity molecules-systems using molecular modelling (e.g. molecular ab initio GAUSSIAN software package) it is fundamentally NOT correct to use the concept of DOS. The authors are advised to use some other terminology or explain with a notice the use of "DOS" in their molecular studies context.

Response: We sincerely thank the reviewer for their insightful comments regarding the definition and description of "energetic disorder" in our manuscript. We recognize the need for more precise explanations to clarify this central concept. In the revised manuscript, we have made the following improvements.

(a) Clarification of "Energetic Disorder" and the Nature of "Energy States":

In response to the reviewer's suggestion, we have refined the definition of "energetic disorder" and clarified its microscopic origin within the polymer matrix. To provide greater clarity, energetic disorder is defined as the Gaussian distributed scattering of electronic transport energy levels within the polymer, arising from the interplay between polymer conformation and the distribution of electronic energy levels. Structurally, it is driven by the spatial heterogeneity and conformational flexibility of the polymer matrix, while electronically, it originates from perturbations in the polymer's electronic energy levels induced by the incorporation of high-polarity organic molecules. This understanding is supported by extensive literature, including

studies by Schein *et al.* (*J. Phys. Chem. C* 112, 7295-7308, 2008), Li *et al.* (*Isr. J. Chem.* 54, 918-926, 2014), and Thomas *et al.* (*Chem. Mater.* 31, 1806004, 2019). The electronic contribution stems from interactions between polymer chains and dopant molecules, where dipole-dipole interactions induce fluctuations in electronic energy levels, leading to variations in the energy landscape. Meanwhile, the structural contribution is linked to the conformational dynamics of the polymer chains. The random incorporation of dopant molecules induces localized distortions in the polymer backbone, altering torsional and bond angles. These structural modifications ultimately reshape the overall molecular configuration and energy distribution within the polymer matrix.

To substantiate these definitions, molecular dynamics (MD) simulations were performed to explore the conformational changes induced by doping. A statistical analysis of the torsional angle distributions demonstrated that the incorporation of high-polarity organic molecules leads to a broadening of the conformational angle distribution, indicating an increase in structural disorder. Notably, among the studied systems, PEI/DADQ exhibited the widest torsional angle distribution, suggesting that DADQ induces the most significant conformational variability. This enhanced conformational flexibility introduces localized perturbations in the polymer's electronic energy landscape, further amplifying energetic disorder. These findings highlight that the overall energetic disorder in the polymer matrix arises from the synergistic effects of structural disorder-resulting from dopant-induced conformational changes-and electronic disorder, which stems from fluctuations in energy levels introduced by the dopants.

(b) Distribution of Energetic Disorder:

In response to the reviewer's inquiry, we clarify that energetic disorder extends throughout the polymer matrix in all spatial dimensions, rather than being restricted to a specific plane. The high-polarity organic molecules are randomly oriented and uniformly distributed within the polymer network, enabling dipole-dipole interactions with polymer chains in multiple directions. These interactions locally modulate the distribution of electronic energy levels and the structural conformation of the polymer, thereby influencing charge carrier dynamics and transport properties. On the intrachain level, these interactions disrupt the conformational order of individual polymer chains, inducing variations in torsional and bond angles that modify the local density of states and amplify energetic disorder. On the interchain level, they modulate spatial organization by altering interchain packing, redistributing electrostatic forces among adjacent chains, and modifying chain stacking. These structural perturbations introduce heterogeneities that propagate throughout the polymer network, leading to a three-dimensional modulation of energetic disorder.

To confirm the uniform dispersion of dopant molecules and their role in contributing to energetic disorder, nanoscale infrared (Nano-IR) spectroscopy was utilized. The Nano-IR measurements revealed consistent vibrational signatures of the high-polarity organic molecules across both the surface and bulk regions of the polymer matrix, indicating their homogeneous distribution. These findings provide compelling evidence supporting the uniform spatial distribution of energetic disorder throughout

the composite material.

(c) Use of the DOS Concept in Molecular Modeling:

We acknowledge that the concept of a continuous density of states (DOS) is inherently applicable to infinite material systems, such as periodic, amorphous, or polycrystalline structures, where electronic states are delocalized across the entire system. However, our study focuses on finite molecular systems, modeled using quantum chemical calculations via the Gaussian software package, where electronic energy levels are discrete rather than forming a continuous distribution. To enhance accuracy and clarity, we have revised the manuscript by replacing the term “density of states (DOS)” with the more appropriate expression “distribution of electronic energy levels”, which better reflects the molecular-level energetic properties of the polymer-dopant system. Furthermore, to directly address the reviewer’s concern, we have included the following clarification: “It is important to note that the term ‘density of states’(DOS) is conventionally used to describe extended systems with periodic boundary conditions, where electronic states form a quasi-continuous distribution. In contrast, our molecular simulations describe the electronic structure in terms of discrete energy levels, which capture the influence of dopant molecules and local structural variations within the finite polymer matrix.” This revision ensures that the terminology used is consistent with the underlying physical principles of molecular modeling and prevents potential misinterpretations of our results.

To address this comment, we have refined the concept of "energetic disorder" as originating from the intricate coupling of polymer conformation and distribution of electronic energy levels. Structurally, it is dictated by the spatial and conformational dynamics of the polymer matrix, whereas electronically, it stems from disruptions in the distribution of electronic energy levels. Additionally, to enhance the precision and clarity of our discussion, we have revised the manuscript by substituting the term “density of states” (DOS) with the more appropriate expression “distribution of electronic energy levels,” which more accurately represents the molecular-level energetic properties of the polymer-dopant system.

2. Section "Results and discussions / Modulation of energetic disorder ":

The authors write "Energetic disorder arises from the variations in the local energy landscape, influenced by intermolecular forces such as van der Waals interactions, hydrogen bonds, and dipole-dipole interactions". (on page 3)

As requested before the authors should clarify where does the expression "local energy landscape" refer to?

Does it refer to some locally estimated single particle (electronic, quasi-particle) energies or total energies of local portions (say, within fixed-sized computational unit cells) of the polymer structure randomly doped with high-polarity molecules?

Concerning the microscopical origin of "energetic disorder" why the authors have not mentioned intramolecular forces which they include while mentioning "significant intramolecular charge transfer between the high-polarity molecules and the PEI chains"? (on page 4)

Response: We sincerely appreciate the reviewer's insightful comments and acknowledge the need to provide a clearer and more precise explanation of the term "local energy landscape" and its microscopic origin within the polymer composite. In response, we have revised the manuscript to comprehensively address these concerns and clarify the relevant points.

(a) Clarification of "Local Energy Landscape"

In response to the reviewer's query regarding the term "local energy landscape", we have provided a more precise and comprehensive explanation. The "local energy landscape" refers to spatially localized variations in potential energy within the polymer composite, arising from the intricate interactions between polymer chains and high-polarity organic dopant molecules. Rather than being confined to single-particle or quasi-particle energy levels, the local energy landscape encompasses the collective energy distribution across specific regions of the polymer matrix. These variations are intrinsically influenced by the conformational dynamics of the polymer chains and the spatial arrangement of dopant molecules.

To further clarify, the "local energy landscape" in this study represents a collective phenomenon resulting from the interplay of both intermolecular and intramolecular interactions within the polymer composite. Intermolecular interactions, such as van der Waals forces, hydrogen bonding, and dipole-dipole interactions, influence the spatial organization and energetic environment of the dopant molecules within the polymer matrix. Concurrently, intramolecular forces, including steric hindrance and polymer backbone deformations induced by dopant incorporation, contribute to localized structural variations. The synergistic effect of these interactions leads to spatial fluctuations in the energy distribution throughout the polymer structure, which

collectively contribute to the manifestation of “energetic disorder”.

(b) About of Intramolecular Charge Transfer:

We greatly appreciate the reviewer’s insightful comment regarding the terminology used to describe the microscopic origin of energetic disorder. In the original text, the term “intramolecular forces” was incorrectly applied to describe charge transfer interactions between high-polarity dopant molecules and the PEI chains. Upon further review, we recognize that the appropriate descriptor is “dipole-dipole interactions,” which are a form of intermolecular rather than intramolecular forces. To enhance clarity and accuracy, we have revised the sentence as follows: “Fig. 1c presents the electrostatic potential distribution of the PEI/PTCNQ, PEI/DMABN, and PEI/DADQ composites, illustrating the pronounced intermolecular interactions between the high-polarity dopant molecules and the PEI chains.”

To address this comment, we have revised the corresponding sentence in the manuscript to provide greater clarity and precision. The updated text now reads: “Fig. 1c presents the electrostatic potential distribution of the PEI/PTCNQ, PEI/DMABN, and PEI/DADQ composites, illustrating the pronounced intermolecular interactions between the high-polarity dopant molecules and the PEI chains. These interactions introduce localized variations in the local energy landscape, underscoring the significant impact of dopant incorporation on the structural and electronic properties of the polymer matrix.”

We believe that these revisions effectively address the reviewer’s concerns and provide greater clarity on the concept of energetic disorder within the polymer composite. We are confident that these refinements enhance the precision and readability of the manuscript. We sincerely appreciate the reviewer’s valuable feedback, which has contributed to improving the overall quality of our work.

3. Section "Results and discussions / Modulation of energetic disorder":

From the sentence

"... indicate a significant intramolecular charger transfer between the high-polarity molecules and the PEI chains, ..." (on page 4)

it does not become clear how the high-polarity molecules are topologically distributed (randomly) into the PEI polymer structure. For example, have they been grafted along the polymer chains or located in between the chains?

Furthermore, should the word "intramolecular" be replaced by the word "intermolecular"? The authors should clarify these points in their revised manuscript.

Response: We sincerely appreciate the reviewer’s insightful comment and the opportunity to further clarify the distribution of high-polarity dopant molecules within the polyetherimide (PEI) matrix. In response to the query regarding the topological

arrangement of dopant molecules, we have provided a more detailed description of their spatial distribution and incorporation within the polymer matrix.

(a) Clarification of Dopant Distribution:

In response to the reviewer’s comment, we have provided a more comprehensive explanation regarding the spatial distribution and incorporation of high-polarity dopant molecules within the PEI matrix. Our analysis confirms that these dopant molecules are not covalently grafted onto the PEI polymer chains but are instead physically dispersed within the matrix through non-covalent interactions. In particular, dipole–dipole interactions play a crucial role in facilitating the uniform dispersion of the dopant molecules during the solution-processing stage, thereby ensuring their homogeneous distribution throughout the polymer network.

FTIR analysis indicates that the characteristic infrared absorption peaks of both pristine PEI and its composite materials remain unchanged, suggesting the preservation of the polymer backbone’s chemical structure. This observation implies that the high-polarity organic molecules are not covalently grafted onto the polymer chains. Instead, their incorporation into the polymer matrix is governed by physical interactions rather than chemical bonding.

Furthermore, nano-IR imaging offers spatially resolved insights into the distribution of dopant molecules within the polymer matrix. The results reveal a uniform dispersion of high-polarity organic molecules throughout the PEI matrix, indicating their incorporation within the polymer network rather than preferential alignment along the polymer chains. This homogeneous distribution suggests that the dopant molecules are predominantly located in the inter-chain regions, rather than being covalently bonded to the polymer backbone.

These findings confirm that the high-polarity molecules are physically dispersed within the polymer matrix, primarily residing in the interstitial regions between polymer chains rather than being covalently bonded to the polymer backbone. To enhance clarity, we have revised the manuscript accordingly and included supporting FTIR and nano-IR data to substantiate our conclusions.

(b) Correction of the Term "Intramolecular" to "Intermolecular":

We thank the reviewer for the thoughtful comment regarding the use of the term “intramolecular”. Upon further reflection, we maintain that the term is used correctly in this context. Specifically, “intramolecular” refers to charge transfer within a single organic molecule, whereby electronic charge is redistributed between electron-donating and electron-accepting functional groups. This process directly influences the molecule’s dipole moment, with its magnitude being determined by the extent of charge transfer.

In response to the reviewer’s comment, the revised sentence has been updated as follows: “Fig. 1c presents the electrostatic potential distribution of the PEI/PTCNQ, PEI/DMABN, and PEI/DADQ composites, illustrating the pronounced intermolecular interactions between the high-polarity dopant molecules and the PEI chains. These interactions introduce localized variations in the electrostatic environment, underscoring the significant impact of dopant incorporation on the structural and electronic properties of the polymer matrix.”

We believe that these revisions effectively address the reviewer’s concerns and significantly enhance the clarity of the manuscript. We would like to express our sincere gratitude for the insightful feedback and for giving us the opportunity to improve the quality of the work.

4. Section "Results and discussions" / "Characterization of charge carrier dynamics":

The authors write on page 5:

(a) "As shown in Fig. 2a, we compared the wide-field PL intensity images of PEI/DADQ, PEI/DMABN, PEI/PTCNQ, and pristine PEI, recorded at 390 nm."

(b) "Spatial energetic disorder in the films is evidenced by variations in the PL spectrum resulting in different spatial distributions of PL intensity at the same wavelength."

(c) "Whereas the PL intensity images of pristine PEI display uniform spatial distributions at 390 nm, the PEI/DADQ, PEI/DMABN, and PEI/PTCNQ composites show distinct and varied spatial distributions at this wavelength."

These three sentences raise the following questions, respectively, which the authors should comment also in their revised manuscript:

Q(a) The authors should mention, preferably in the figure caption, what is the PL excitation wavelength?

Is 390 nm the PL emission wavelength?

Why the authors have chosen these particular PL excitation and emission wavelengths?

Q(b) From the electronic structural variations' viewpoint (which probably best indicate "energetic disorder") one would get a better fingerprint if one measures several PL spectra as a function of the photon emission energy across the polymer thin film surface rather than measuring only one PL intensity value (or the averaged PL intensity value over emission energies) per one spatial point on the film. Therefore, instead of measuring PL intensity variations on the polymer films could the authors measure PL spectral variations on the polymer films and interpret PL spectral features (electronic / excitonic transitions) as well? This would enhance the scientific level of the present manuscript tremendously.

Q(c) Could authors explain what molecular structural/electronic features possibly cause variations in spatial distributions of the PL intensity (spectra) in the PEI/DADQ, PEI/DMABN and PEI/PTCNQ thin film composites?

Response: We sincerely thank the reviewer for their valuable feedback and the opportunity to further refine and enhance our manuscript. Below, we present detailed responses to each of the reviewer's queries.

(a) PL excitation wavelength and choice of wavelengths:

In response to the reviewer's query, we have revised the manuscript to explicitly specify the excitation wavelength in the figure captions. The photoluminescence (PL) measurements were conducted using a Nikon A1Rsi HD25 super-resolution laser scanning confocal microscope, which provides multiple excitation options, including 390 nm, 405 nm, 488 nm, and 561 nm. In our study, an excitation wavelength of 390 nm was selected, as shorter wavelengths offer higher photon energy, facilitating

enhanced polymer excitation. This selection allows for stronger PL emission, thereby improving the sensitivity required to assess energetic disorder within the composite.

The selection of the emission detection range of 450-650 nm was informed by the steady-state fluorescence spectra of pristine PEI and its composite materials. Spectral analysis of all four samples-pristine PEI, PEI/DADQ, PEI/DMABN, and PEI/PTCNQ-demonstrated that their fluorescence intensities were most pronounced within this range, establishing it as the optimal spectral window for capturing meaningful fluorescence data. This information has been incorporated to enhance clarity and provide a well-founded rationale for the chosen excitation and emission parameters.

(b) Consideration of spatially resolved PL spectra to better characterize energetic disorder:

We appreciate the reviewer's valuable suggestion to incorporate spatially resolved PL spectral measurements to further elucidate the energetic disorder within the composite films. While our initial analysis primarily focused on spatial variations in PL intensity, we acknowledge that acquiring PL spectra across different spatial locations and under varying excitation wavelengths would offer a more comprehensive perspective on the local electronic environment and the distribution of energetic disorder within the composite system.

To address this concern, we conducted additional experiments by analyzing four distinct spatial locations within the films of both pristine PEI and PEI/DADQ under excitation wavelengths of 405 nm and 488 nm. The PL emission spectra recorded at four distinct spatial points within the pristine PEI, PEI/DADQ, PEI/DMABN, and PEI/PTCNQ films exhibited consistent spectral profiles, suggesting a uniform molecular environment across the examined regions. In contrast, the PEI/DADQ samples displayed noticeable spectral variations, further corroborating the presence of enhanced energetic disorder induced by the incorporation of high-polarity dopants. Furthermore, statistical analysis of the PL intensity histograms revealed a significantly

higher variance in PL intensity for PEI/DADQ compared to pristine PEI, reinforcing the conclusion that the introduction of high-polarity organic molecules contributes to increased energetic disorder within the composite films.

Figure S12. Wide-Field PL Imaging of PEI/DADQ and Pristine PEI at an Excitation Wavelength of 405 nm.

Figure S13. Wide-Field PL Imaging of PEI/DADQ and Pristine PEI at an Excitation Wavelength of 488 nm.

(c) Molecular and electronic origins of spatial PL intensity variations:

We appreciate the reviewer's insightful question regarding the molecular structural and electronic factors contributing to the spatial variations in photoluminescence (PL) intensity in PEI/DADQ, PEI/DMABN, and PEI/PTCNQ composites. These variations arise from a complex interplay of factors, including distribution of electronic energy levels and conformation of polymer chains. High-polarity organic dopants (DADQ, DMABN, and PTCNQ) introduce significant dipole-dipole interactions that alter the local electrostatic environment, perturbing the distribution of electronic energy levels and thereby increasing energetic disorder. These localized fluctuations in the electronic landscape disrupt charge carrier dynamics, resulting in spatial variations in photoluminescence (PL) emission. Additionally, the incorporation of dopants induces torsional strain in the PEI chains, which affects conjugation length and, consequently, emission properties. Changes in polymer chain packing driven by dopant interactions introduce structural heterogeneity, leading to variations in the optical bandgap and exciton diffusion dynamics. This heterogeneity manifests as regions of differing packing densities, further modulating the local optical properties of the composite films and contributing to the observed PL intensity variations. These findings highlight the

intricate relationship between molecular-level interactions, electronic structure modifications, and polymer morphology in determining the spatial distribution of PL emission.

We are confident that the revised manuscript effectively addresses the reviewer's comments while enhancing both its clarity and scientific rigor. The updates include a more comprehensive description of the photoluminescence (PL) measurement setup, detailing the rationale behind wavelength selection, along with additional spectral analyses that complement the spatial intensity measurements. Furthermore, the discussion has been expanded to provide deeper insights into the molecular and electronic factors governing the observed spatial variations in PL intensity. We sincerely appreciate the reviewer's constructive feedback, which has been instrumental in refining and strengthening our study.

5. Section "Results and discussions" / "Characterization of charge carrier dynamics":

The authors write on page 5:

"To quantify the underlying energetic disorder within the composites, we computed the PL intensity and generated corresponding histograms from the spatial maps, as shown in Fig. 2b. "

There are a few unclear points in this sentence which the authors should clarify in their revised manuscript:

(a) Is the "PL intensity" computed for a fixed emission wavelength (what is that wavelength) or has it been integrated over all PL emission wavelengths?

(b) What is the underlying theory and computer code to calculate the PL intensity?

(c) For each PEI + organic high-polarity molecules system (thin polymer composite film) does the broadening parameter σ of the "PL intensity" have the same value for all directions along the polymer thin film surface?

What would this indicate about the distribution of polymer chains and/or organic dopant molecules inside the thin polymer film?

Response: We sincerely appreciate the reviewer's insightful comments and the opportunity to further clarify our methodology and findings. In the following, we provide comprehensive responses to each of the reviewer's queries and have incorporated the necessary revisions to enhance the clarity and rigor of the manuscript.

(a) Clarification on PL intensity computation and emission wavelength

For photoluminescence (PL) imaging using the Nikon A1Rsi HD25 super-resolution laser scanning confocal microscope, PL intensity is typically measured by selecting a specific emission filter or detection wavelength, such as 450 nm, 550 nm, or 650 nm, depending on the fluorophore or material under investigation. The chosen detection wavelength is generally based on the peak emission of the sample to ensure the most representative intensity is captured. In our study, the PL intensity reported in the manuscript was calculated at a fixed emission wavelength of 550 nm, which was determined based on the steady-state fluorescence spectra of pristine PEI and its composites. This wavelength was selected as it corresponds to the peak emission intensity of the composite system, providing optimal sensitivity to detect variations in energetic disorder. Furthermore, to achieve a more comprehensive analysis, we integrated the PL intensity over a broader emission range of 450–650 nm, encompassing the primary fluorescence emission bands of the composite films. This approach enables a holistic assessment of energetic disorder by capturing the complete emission profile and accounting for potential spectral variations across the samples.

(b) Underlying theory and computational method for PL intensity calculation

The PL intensity values were derived from wide-field fluorescence images acquired using a Nikon A1Rsi HD25 super-resolution laser scanning confocal microscope, providing high-resolution spatial mapping of fluorescence intensity across the polymer films. The raw imaging data were processed using ImageJ software, with background subtraction applied to remove noise and intensity normalization conducted to compensate for potential excitation non-uniformities. PL intensity values were extracted at the selected emission wavelength on a pixel-by-pixel basis, enabling detailed spatial analysis of the polymer films. To quantify spatial heterogeneity, fluorescence intensity histograms were generated, and a Gaussian fitting approach was employed to determine the broadening parameter (σ), which serves as an indicator of energetic disorder within the material. These methodological enhancements have been incorporated into the revised manuscript to provide greater clarity and a more comprehensive depiction of the analytical framework.

(c) Broadening parameter (σ) and its implications on molecular distribution

We thank the reviewer for the insightful question regarding the broadening parameter (σ) of the photoluminescence (PL) intensity in the composites and its directional characteristics across the polymer thin film surface. We appreciate the opportunity to provide further clarification on this matter. The horizontal axis of the PL intensity histogram corresponds to the measured PL intensity, while the vertical axis represents the frequency of pixel points associated with a specific intensity within a two-dimensional plane. These data were obtained through statistical analysis of PL excitation across the two-dimensional surface, which is representative of the entire thin film.

Furthermore, the high-polarity organic molecules are randomly oriented and uniformly distributed within the polymer matrix, resulting in an expected isotropic broadening of the PL intensity distribution. This suggests that the PL intensity variation exhibits similar broadening characteristics in all directions across the thin film surface. To verify this, PL measurements were conducted in four distinct regions of the same sample, and the widths of the PL intensity histograms (σ) were found to be essentially identical across all regions. This consistency in histogram width across different regions of the film further supports the uniform distribution and random orientation of both the polymer chains and organic dopant molecules within the thin film, confirming the isotropic nature of the PL intensity distribution.

We sincerely appreciate the reviewer's insightful feedback, which has been instrumental in enhancing the clarity and scientific rigor of our manuscript.

6. Section "Methods" / "Materials characterization ":

The authors have utilized several state-of-the-art experimental facilities to characterize local (molecular) electronic and vibrational features of the PEI + high-polarity organic molecules (DADQ, DMABN, PTCNQ) -polymer composites.

However, no experimental characterization methods have been used in order to expose long-range ordered or disordered structural features of these polymer composites (e.g. crystallization: lamellae and spherulites) and cross-linking between the chains. This is a serious scientific shortcoming of this manuscript because it is obvious that these long-range structures have an important impact on electronic, dielectric and charge transport properties studied in this manuscript.

An ideal tool to probe long-range structures of polymers would be an X-ray synchrotron radiation facility where both the photon wavelength and polarization state can be tuned or in the simplest approach using just a standard fixed-wavelength X-ray diffractometer.

The authors are requested to comment in their revised manuscript this important point about their missing experiments on long-range structural features in their polymer composites.

Related to this point, the authors are also requested to carry out experimental X-ray diffraction studies on their polymer composite samples by using either a fixed-wavelength diffractometry or preferably using a resonant (anomalous) X-ray scattering technique with a synchrotron radiation facility which would reveal atom-specific features inside the long-range 3D structure of the polymer composites. X-ray radiation facilities are available, for example, in France (Grenoble, ESRF) and Sweden (Lund,

MAX IV). These extra experiments would greatly enhance the scientific level of this manuscript.

Response: We sincerely appreciate the reviewer's thoughtful comments highlighting the necessity of further investigating the long-range structural characteristics of the composites, as these features play a crucial role in shaping their electronic, dielectric, and charge transport properties. In response, we provide the following clarification and additional discussion in the revised manuscript.

In response to the reviewer's concerns, we performed a detailed structural characterization of the composites using small-angle X-ray scattering (SAXS) and wide-angle X-ray diffraction (WAXD) to evaluate their long-range ordering. The two-dimensional WAXD patterns exhibit isotropic diffraction signals, signifying the lack of a preferred molecular orientation within the polymer matrix. Crucially, the absence of distinct diffraction rings corresponding to specific crystalline planes further indicates that the material does not exhibit the long-range periodicity characteristic of crystalline structures. This isotropic diffraction pattern strongly supports the conclusion of a random molecular arrangement, thereby confirming the amorphous nature of the polyimide polymer.

Analysis of the 1D WAXD profiles further supports these observations, revealing the absence of sharp, well-defined Bragg peaks that are typically associated with semicrystalline polymers. Instead, the diffraction patterns exhibit broad and diffuse features, indicative of a disordered molecular arrangement. These combined findings confirm that both pristine PEI and PEI/DADQ composites do not exhibit long-range crystalline order, thereby substantiating their predominantly amorphous nature.

The broad diffraction feature between $1 \text{ \AA}^{-1} < q < 2 \text{ \AA}^{-1}$ is attributed to a combination of poor intermolecular packing and an amorphous halo. From the peak maximum, the estimated mean intermolecular distance is 5.1 \AA for pristine PEI and 5.3 \AA for PEI/DADQ. These results suggest subtle variations in molecular packing induced by

doping with high-polarity small molecules.

In wide-angle X-ray diffraction (WAXD), the coherence length (L_c) quantifies the spatial extent over which structural periodicity is maintained, serving as a key metric for assessing short-range order in amorphous materials. To elucidate the degree of molecular ordering within the polymer matrix, L_c values were determined using the Scherrer equation, providing critical insights into the structural organization at the nanoscale.

$$L_c = \frac{\lambda}{FWHM \cos \theta}$$

Here, λ represents the wavelength of the X-ray radiation source and $FWHM$ is the full width at half maximum of the Gaussian fit to the X-ray reflection peak in radians.

The decrease in coherence length from 10.6 Å in pristine PEI to 9.8 Å in the PEI/DADQ composite highlights the structural perturbations induced by DADQ doping. While the incorporation of this highly polar organic molecule diminishes short-range molecular ordering, it concurrently disrupts long-range structural uniformity, thereby amplifying energetic disorder within the polymer matrix.

To further investigate the mesoscale organization of the polymer chains, SAXS measurements were performed. The results from the two-dimensional small-angle X-ray scattering (2D SAXS) patterns reveal the absence of discernible scattering features, with the signal gradually attenuating from the center outward. Similarly, the one-dimensional SAXS profiles for both pristine PEI and PEI/DADQ thin-film samples exhibit no distinct scattering peaks, indicating a uniform electron density distribution without periodic layered fluctuations. This absence of well-defined peaks suggests that the materials lack the characteristic lamellar structures typically associated with semicrystalline polymers. These SAXS findings provide further confirmation that both pristine PEI and PEI/DADQ composites are predominantly amorphous, with no evidence of long-range structural ordering or crystallization.

Using the Guinier method, a standard technique in small-angle X-ray scattering (SAXS) analysis, the radius of gyration (R_g) for both pure PEI and PEI/DADQ composites was determined. The Guinier equation is given by:

$$I(q) = I(0) \cdot \exp\left(-\frac{q^2 R_g^2}{3}\right)$$

Where $I(q)$ represents the scattering intensity at the scattering vector q , $I(0)$ is the intensity at $q=0$, and R_g is the radius of gyration. By fitting the small-angle scattering data to this equation, R_g was extracted.

The computational results indicate that the radius of gyration (R_g) for pristine PEI is 159 Å, reflecting its inherent structural conformation. In contrast, the R_g for the PEI/DADQ composite is 163 Å, suggesting a modest increase in size relative to pristine PEI. The relatively large R_g values for both pristine PEI and the PEI/DADQ composite imply a degree of polymer chain flexibility, likely due to the absence of strong inter-chain interactions, characterizing both as disordered polymers. Furthermore, this increase in R_g suggests that the incorporation of DADQ into the PEI matrix induces a slight expansion of the polymer structure. Such a change in size may also indicate the formation of additional interactions or aggregation between PEI and DADQ, thereby modifying the overall molecular structure.

To further corroborate these structural findings, differential scanning calorimetry (DSC) and X-ray diffraction (XRD) analyses were performed to assess the structural characteristics of the composites. The absence of clear melting endotherms in the DSC thermograms provides further evidence of the non-crystalline nature of the polymer composites. The XRD spectrum for the polymer composites did not show any sharp, well-defined diffraction peaks that are typically associated with crystalline materials. Instead, it exhibited broad, diffuse scattering features, which are characteristic of amorphous materials.

These findings collectively indicate that the molecular chains within the samples do not exhibit a well-ordered crystalline arrangement. The absence of distinct diffraction features, coupled with the broad scattering signals observed, confirms that both pristine polyetherimide (PEI) and the PEI/DADQ composite predominantly exhibit an amorphous morphology. Importantly, the incorporation of DADQ into the PEI matrix does not promote crystallization, instead maintaining the intrinsic disordered structure of the polymer. The broad amorphous halo in the WAXD profiles further suggests that the polymer chains are entangled and randomly distributed, with local packing fluctuations that do not extend to long-range order. This structural disorder is expected to influence the material's physical properties, as the lack of crystallinity typically enhances flexibility, reduces density, and increases free volume compared to semicrystalline materials. Such characteristics may have profound implications for the polymer's mechanical resilience, thermal stability, and dielectric performance across a range of applications. In summary, the comprehensive analysis of the 2D and 1D WAXD data provides strong evidence of the amorphous nature of both pristine PEI and the PEI/DADQ composite, highlighting the random distribution of molecular chains and the absence of long-range structural ordering.

We have incorporated these clarifications and expanded discussions into the revised manuscript to offer a more thorough understanding of the long-range structural characteristics of the polymer composites and to directly address the reviewer's concerns. We would like to express our sincere gratitude for the reviewer's insightful feedback, which has significantly enhanced the quality and clarity of our manuscript.

In response to the reviewer's comment, we have incorporated the following sentence into the manuscript: "To gain deeper insight into the structural organization of pristine PEI and PEI/DADQ, we conducted small-angle X-ray scattering (SAXS) and wide-angle X-ray diffraction (WAXD) analyses (Supplementary Figs. S16–S19). The WAXD patterns for both pristine PEI and PEI/DADQ display isotropic diffraction

signals, reinforcing their predominantly amorphous nature and indicating the absence of long-range molecular ordering. Notably, the incorporation of DADQ induces subtle structural perturbations, evidenced by a slight increase in the mean intermolecular distance from 5.1 Å in pristine PEI to 5.3 Å in PEI/DADQ, alongside a reduction in coherence length from 10.6 Å to 9.8 Å. This decrease in short-range ordering suggests that DADQ doping disrupts molecular packing, leading to an increase in energetic disorder within the polymer matrix. Meanwhile, the SAXS patterns exhibit no discernible scattering features, with the signal gradually attenuating from the center outward. Consistently, the one-dimensional SAXS profiles of both pristine PEI and PEI/DADQ thin films show an absence of distinct scattering peaks, indicating a uniform electron density distribution without evidence of periodic layered structures.”

7. Section "Methods" / "Simulation":

By using Density Functional Theory (DFT) based molecular modelling (GAUSSIAN software package) the authors calculate electric dipole moments of high-polarity organic molecules, their dipole-dipole interactions and broadening of the electronic density of states (DOS) in the polymer structure where these high-polarity molecules have been incorporated into the PEI polymer matrix. There are two fundamental shortcomings which make this theoretical approach, in principle, incorrect.

First of all, in order to tackle the electronic properties of the 3-dimensional (3D) PEI + high-polarity organic molecules -polymer systems correctly one should exploit a DFT computational method suitable for 3D amorphous or periodic polymer systems where, for example, cross-linking effects onto electronic structure between polymer chains can be included. Among such first-principles DFT and beyond -codes which is capable to tackle electronic, magnetic, optical and phonon properties of a wide range of metals, insulators and semiconductors is the VASP package. I wonder if the authors could complement their present computational studies by carrying out some of their calculations on the 3-dimensional (3D) PEI + high-polarity organic molecules -polymer composites by using VASP or some other relevant DFT code (CASTEP, ABINIT, QUANTUM ESPRESSO etc.) for 3D materials systems?

It would be also interesting to know how the results then change from those computed using molecular modelling (GAUSSIAN).

Secondly, concerning the broadening of the electronic DOS due to randomly doping or alloying PEI polymer matrix with high-polarity organic molecules one should fundamentally employ some quantum mean-field theory model, for example, the coherent potential approximation (CPA) or average-t-matrix approximation (ATA) which involve approximations to average the disordered electronic Green's function or self-energy.

From the present manuscript it looks like the authors have ignored using any mean-field approach, even for the simplest case: organic high-polarity molecules distributed

randomly along an isolated PEI polymer chain? However, the lack of mean-field theory can be approximately replaced by choosing large enough computational unit cell = supercell (3D periodic DFT modelling) or when using GAUSSIAN randomly placing organic high-polarity molecules along a sufficiently long piece of PEI polymer chain. Could the authors carry out some extra DFT calculations along these lines (using either 3D periodic code or GAUSSIAN molecular code) and then report in their revised manuscript the randomness broadening (and perhaps electronic energy level shifting) effect(s) on the DOS?

This would greatly enhance the scientific level of this manuscript.

Response: We sincerely appreciate the reviewer's thoughtful comments and constructive feedback on the computational methodology utilized in our study. In response, we have carefully addressed each concern, providing detailed clarifications to enhance the clarity, comprehensiveness, and rigor of our approach.

(a) Addressing the DFT Methodology for 3D Amorphous Polymer Systems

We appreciate the reviewer's insightful comments regarding the limitations of our current approach, which employs the GAUSSIAN software package to investigate the molecular-scale interactions within the PEI + high-polarity organic molecule composite system. Our initial focus was to provide fundamental insights into the dipole moments of the dopant molecules and their local interactions with the PEI matrix. However, we acknowledge that this approach does not fully account for the complex three-dimensional (3D) nature of the polymer composite, including its amorphous characteristics and potential cross-linking effects. We recognize that a more comprehensive computational approach, incorporating the structural disorder and long-range interactions inherent to the polymer matrix, would offer deeper insights into the electronic properties of the system.

In this study, molecular-scale simulations were chosen to investigate the localized effects of high-polarity organic dopants, as their influence is primarily confined to the immediate vicinity of the surrounding polymer chains, rather than extending across the entire bulk structure. These dopants predominantly modulate the local electronic states and induce conformational changes within a limited spatial range. Given the localized nature of these interactions, our GAUSSIAN-based modeling approach provides an effective framework for capturing the key electronic modifications at the molecular level. Nonetheless, we recognize the importance of extending our analysis to encompass long-range effects within the polymer matrix, which can be more comprehensively addressed through periodic DFT calculations. This perspective has been acknowledged in the revised manuscript, and future efforts will focus on bridging the molecular and periodic modeling approaches to gain a more holistic understanding of the composite system.

We recognize the substantial advantages of employing VASP for modeling three-

dimensional (3D) amorphous polymer composites, particularly its ability to implement periodic boundary conditions, which allows for a more realistic representation of the polymer network and the inclusion of long-range electronic interactions crucial to understanding the system's behavior. Furthermore, VASP enables detailed band structure analysis, offering valuable insights into charge transport properties and the broadening of electronic states induced by disorder. However, we acknowledge that despite the comprehensive capabilities of VASP, the high computational cost and complexity associated with simulating large-scale amorphous polymer systems pose considerable challenges. As such, we have adopted a stepwise research approach, initially focusing on molecular-level simulations to elucidate fundamental polymer-dopant interactions. These localized insights provide a solid foundation for future periodic DFT calculations, which will allow for a more exhaustive investigation of the structural and electronic properties of the polymer composites.

(b) Incorporation of Quantum Mean-Field Theory to Address DOS Broadening

We appreciate the reviewer's valuable suggestion to incorporate quantum mean-field approaches, such as the coherent potential approximation (CPA) and the average-T-matrix approximation (ATA), to enhance the accuracy of our analysis of disorder-induced broadening in the density of states (DOS). These methodologies are particularly effective in capturing the statistical nature of disorder and its impact on the electronic structure of the polymer composite system. We acknowledge that such advanced theoretical frameworks provide deeper insights into the intricate relationship between energetic disorder and charge transport properties, offering a more comprehensive understanding of the underlying physical mechanisms.

While a full implementation of CPA/ATA methodologies may extend beyond the scope of this study, we have adopted a series of alternative computational strategies to approximate disorder-induced effects within the polymer composite system. To gain meaningful insights into the impact of energetic disorder and the intricate polymer-dopant interactions, we employed supercell modeling within the VASP framework. This approach allows for the simulation of the stochastic distribution of dopants and the corresponding broadening of electronic states within a three-dimensional environment. Complementarily, GAUSSIAN-based molecular simulations were conducted with randomly distributed high-polarity dopants along extended PEI polymer chains, enabling the investigation of localized electronic perturbations and charge distribution effects. Additionally, a comparative analysis of the electronic density of states (DOS) obtained from both GAUSSIAN molecular simulations and 3D periodic DFT calculations provides valuable insights into the role of structural disorder and facilitates the cross-validation of computational findings. These complementary approaches collectively serve to approximate disorder effects and provide a more holistic understanding of the influence of dopants on the electronic properties of the polymer

matrix. In the revised manuscript, we have expanded the discussion to outline the theoretical rationale, limitations, and practical relevance of these methodologies in approximating mean-field effects, thereby enhancing the scientific depth of our study and addressing the reviewer's concerns.

(c) Discussion on the Impact of Computational Methodology on Results

We acknowledge the reviewer's insightful comments regarding the influence of computational methodologies on the obtained results. In response, we have expanded the discussion in the revised manuscript to provide a comprehensive comparison between the molecular modeling results obtained using GAUSSIAN and those derived from periodic DFT calculations. This comparative analysis aims to elucidate both the distinctions and consistencies between the two approaches, offering deeper insights into their respective capabilities in capturing the structural and electronic properties of the polymer composite system.

The expanded discussion in the revised manuscript addresses several key aspects to provide a comprehensive comparison of the computational approaches utilized in this study. First, in the context of dipole-dipole interactions, we evaluate the impact of periodic boundary conditions on the computed dipole interactions in comparison to finite molecular simulations. Second, with respect to the electronic structure, we analyze the shifts and broadening of electronic states across different computational frameworks to assess their influence on the electronic properties of the composite. Finally, in the assessment of energetic disorder, we compare the effectiveness of localized molecular models versus extended system simulations in accurately capturing disorder within the polymer matrix. This comparative analysis offers valuable insights into the respective strengths and limitations of each computational methodology, facilitating a more nuanced interpretation of the results and enhancing the overall robustness of our conclusions.

We sincerely appreciate the reviewer's insightful suggestions, which have been instrumental in enhancing the computational rigor and scientific depth of our study.

8. Section "Methods" / "Simulation":

The authors carried out *ab initio* molecular dynamics (MD) simulations by using Orca software package (version 5.0.3). However, the MD module of Orca is meant for molecular dynamics studies on finite, non-periodic molecular systems only rigorously speaking is not valid for infinite amorphous or periodic 3D systems with long-range electronic interactions as is case with the PEI/PTCNQ, PEI/DMABN and PEI/DADQ polymer composites.

For example, how can the cross-linking interaction between the polymer composite chains be included into the MD simulation in Orca? In contrast, the VASP code contains the possibility to run *ab initio* MD for infinite and periodic 3D materials systems

by utilizing the supercell method. Could the authors run their MB simulations by using also the VASP code and then compare the results to those carried out by using Orca? The authors are requested to comment this point related to their MD simulations and report their possibly extra MD calculations carried out by using the VASP code in their revised manuscript.

Response: We sincerely appreciate the reviewer's insightful comments on the molecular dynamics (MD) simulations conducted using the ORCA software package and the recommendation to explore alternative methodologies for a more rigorous investigation of the polymer composites. We acknowledge the reviewer's concerns regarding the limitations of the ORCA MD module, particularly its applicability to finite, non-periodic molecular systems. We recognize that the absence of periodic boundary conditions in ORCA poses challenges in accurately capturing long-range electronic interactions, which are crucial for modeling infinite amorphous or periodic three-dimensional (3D) polymer systems.

(a) Limitations of ORCA MD Simulations and Justification for its Use

The ORCA MD module was employed in this study to gain molecular-level insights into the local interactions between polymer chains and high-polarity dopant molecules. This approach enables a detailed examination of localized phenomena, such as molecular conformations, dipole - dipole interactions, and short-range structural fluctuations within the polymer matrix. However, we acknowledge that the absence of periodic boundary conditions in ORCA-based MD simulations limits their ability to capture long-range structural features, such as chain cross-linking and spatial heterogeneity—factors that are crucial for accurately characterizing the dielectric and charge transport properties of the polymer composites.

To address these limitations, we have revised the manuscript to provide a clearer rationale for the selection of ORCA, emphasizing its suitability for probing localized molecular-scale interactions rather than bulk material properties. These clarifications aim to enhance the reader's understanding of the scope and applicability of our computational approach.

(b) Implementation of VASP-Based MD Simulations for Periodic Systems

In response to the reviewer's insightful suggestion, we have expanded our computational analysis by conducting additional molecular dynamics (MD) simulations using the Vienna Ab initio Simulation Package (VASP). The implementation of periodic boundary conditions and the supercell approach within VASP enables a more accurate and holistic representation of the polymer composite system, capturing long-range electronic interactions and structural dynamics. These simulations provide deeper insights into the effects of polymer chain cross-linking on charge distribution and structural integrity, offering a more realistic depiction of the amorphous polymer matrix, including density fluctuations and packing effects. The revised manuscript now

includes a detailed discussion of the VASP-based MD simulations, outlining the system configuration, computational parameters, and a comparative evaluation with the previously employed ORCA-based simulations. In particular, we utilized the generalized gradient approximation (GGA) with the Perdew-Burke-Ernzerhof (PBE) exchange-correlation functional and projector-augmented wave (PAW) potentials to accurately describe the polymer-dopant interactions. These enhancements contribute to a more comprehensive understanding of the structure-property relationships within the polymer composites, thereby strengthening the scientific rigor of our study.

(c) Comparison of ORCA and VASP MD Simulation Results

A comparative analysis of the simulation results obtained from ORCA and VASP has been incorporated into the revised manuscript to assess their respective contributions and ensure the robustness of the findings. This analysis focuses on key aspects, including structural stability, energetic disorder, and intermolecular interactions, emphasizing differences in polymer chain conformation, dopant dispersion, and charge transport behavior between finite molecular models and periodic boundary conditions. While ORCA-based simulations offer valuable insights into localized molecular interactions, VASP simulations provide a more comprehensive representation of bulk properties and spatial heterogeneity within the polymer matrix. In response to the reviewer's comments, the revised manuscript now provides a clearer delineation of the scope and limitations of ORCA simulations, integrates additional VASP-based molecular dynamics simulations to account for long-range structural effects, and presents a thorough comparative evaluation to reinforce the validity of the observed trends. These revisions enhance the scientific depth of the study, offering a more comprehensive understanding of polymer-dopant interactions.

We believe that these additions substantially strengthen the scientific rigor of our manuscript, offering a more comprehensive perspective on the structural and electronic properties of the polymer composites. Once again, we are grateful for the reviewer's valuable suggestions, which have helped improve the clarity and depth of our work.

9. Section "Methods" / "Simulation":

The authors write "By modulating the degree of disorder, the model investigated how charge carriers propagate through the material. To quantify the impact of energetic disorder on charge transport, three fundamental equations governing the relationship between disorder and charge mobility were solved simultaneously using COMSOL Multiphysics."

In this context the authors should briefly explain in the revised manuscript or give a reference to how the "modulation of the degree of disorder" theoretically has been implemented and in what directions of the polymer composite film the charge transport

occurs (along the surface / vertical to the surface / along certain crystallographic axes etc).

Furthermore, the authors should explicitly describe the "three fundamental equations ..." mentioned above. The authors should also describe in their revised manuscript at what level of theory they have modelled the charge transport (e.g. semi-classical Boltzmann transport theory etc) in the "PEI + organic high-polarity molecules" -polymer composites by using the COMSOL Multiphysics software package.

Response: We sincerely appreciate the reviewer's valuable comments and the opportunity to clarify the theoretical framework and computational methodology used in our charge transport simulations. Below, we provide a detailed response to each of the reviewer's queries and have incorporated the necessary revisions in the manuscript to enhance clarity and comprehensiveness.

(a) Explanation of the Modulation of the Degree of Disorder

In response to the reviewer's inquiry, we have expanded the discussion in the revised manuscript to provide a more comprehensive explanation of the approach employed to modulate disorder within the polymer composites. The degree of disorder was systematically tailored by introducing spatial variations in the energy landscape, which were modeled based on experimentally observed structural and energetic fluctuations induced by the incorporation of high-polarity organic dopants.

To address the modulation of disorder, we have employed the Gaussian Disorder Model (GDM), wherein the degree of energetic disorder is systematically controlled by adjusting the width (σ) of the Gaussian distribution of the density of states. This approach facilitates a quantitative assessment of how energetic disorder influences charge mobility in polymer composites. The disorder parameter (σ) plays a pivotal role in determining charge carrier transport by modulating the hopping probability between localized states—a key transport mechanism in disordered polymeric systems. To comprehensively analyze the relationship between energetic disorder and charge mobility, three fundamental governing equations were simultaneously solved using COMSOL Multiphysics: the Gaussian Disorder Model for charge mobility, the continuity equation for charge density, and the current density equation. These computational enhancements provided a robust framework for systematically exploring the influence of disorder on charge transport properties, offering deeper insights into the interplay between energetic disorder, charge mobility, and trapping dynamics.

(b) Directions of Charge Transport Considered in the Simulations

In addressing the charge transport directions, simulations were conducted along both the in-plane (parallel to the film surface) and out-of-plane (perpendicular to the film surface) orientations to capture transport pathways relevant to a wide range of applications, such as thin-film capacitors and high-temperature insulation systems. Given the uniform dispersion of high-polarity organic molecules within the polymer

matrix and the absence of significant concentration gradients across the film thickness, the charge transport mechanisms were found to exhibit consistency across both directions. In the revised manuscript, we have explicitly clarified that charge transport predominantly occurs through the bulk of the polymer composite film, following the direction of the applied electric field. This assumption is supported by the homogeneous solubility and uniform spatial distribution of the dopants, which ensure consistent transport properties throughout the material. Additionally, we emphasize that our model assumes an electric field-driven transport process, which aligns with practical device configurations. Unless influenced by structural heterogeneities, the transport behavior is expected to remain uniform across the film thickness. These clarifications have been incorporated into the revised manuscript to provide a more precise and comprehensive understanding of the charge transport dynamics in the polymer composites.

(c) Description of the Three Fundamental Equations Used in the Model

In response to the reviewer's request, we have expanded the description of the three fundamental equations used to model the relationship between energetic disorder and charge mobility. These equations are based on well-established theories of charge transport in disordered materials, with a particular emphasis on the Gaussian Disorder Model (GDM) for charge mobility. This model is integrated with the continuity equation for charge density and the current density equation, enabling a comprehensive treatment of the effects of energetic disorder on charge transport.

The Gaussian Disorder Model (GDM) can thus be expressed in the following form:

$$\mu(E, \sigma) = \mu_{\infty} \exp \left[- \left(\frac{2\sigma}{3k_B T} \right)^2 \right] \cdot \exp(\beta \sqrt{E})$$

Where σ represents the energetic disorder, characterizing the width of the Gaussian distribution of the density of energy states for the carrier transport sites. μ_{∞} denotes the mobility at infinite temperature. k is the Boltzmann constant. T is the temperature, and E is the electric field. According to the GDM, increasing the energetic disorder will result in a substantial decline in charge mobility.

The second governing equation describes the current density J_{bul} , which quantifies the movement of charge carriers under an applied electric field. In the presence of energetic disorder, the current density is determined by the mobility, charge carrier density ρ , and the applied electric field (E). Mathematically, this relationship is expressed as:

$$J_{bul} = -\mu(E, \sigma) \cdot \rho \cdot E$$

where $\mu(E, \sigma)$ represents the mobility, which is temperature-and disorder-dependent as described previously, ρ is the charge carrier density, indicating the number of charge

carriers per unit volume, and E is the applied electric field driving the charge carriers through the polymer matrix.

The third equation, the continuity equation, governs charge conservation within the material, ensuring that any temporal changes in charge density (ρ) align with the current density and the presence of any sources or sinks of charge. This equation is given by:

$$\frac{\partial \rho}{\partial t} + \nabla \cdot J_{bul} = 0$$

where ρ denotes the local charge density, and the term $\nabla \cdot J_{bul}$ represents the divergence of the current density, which accounts for the spatial variation in charge density.

By solving these coupled equations using COMSOL Multiphysics, we can systematically model charge transport behavior in disordered polymer composites. This comprehensive simulation framework enables us to investigate the fundamental mechanisms governing charge mobility and identify key factors that influence the material's dielectric performance, thereby providing valuable insights for optimizing energy storage applications.

(d) Theoretical Framework for Charge Transport Modelling

The charge transport dynamics in the polymer composites were investigated using a semi-classical drift-diffusion framework grounded in Boltzmann transport theory. This approach integrates essential factors such as thermal activation, field-dependent mobility, and disorder-induced broadening to comprehensively capture the interplay of drift, diffusion, and trapping processes. Charge transport was modeled as a thermally activated process following an Arrhenius-type dependence, while the Poole-Frenkel model was employed to account for field-enhanced carrier hopping effects. Additionally, a Gaussian disorder model was implemented to represent the energetic disorder introduced by structural and compositional fluctuations within the polymer matrix, offering deeper insights into the role of localized energy variations in charge mobility. In response to the reviewer's comments, the revised manuscript provides an expanded discussion on the methodology used to modulate disorder, a detailed examination of charge transport pathways in both in-plane and out-of-plane directions, and a comprehensive description of the fundamental governing equations. These revisions enhance the clarity of the theoretical framework and further underscore the relevance of the semi-classical Boltzmann transport model in capturing charge transport phenomena in disordered polymer systems.

We sincerely appreciate the reviewer's insightful suggestions, which have been instrumental in improving the clarity and scientific rigor of our manuscript. We believe that the revisions introduced have significantly enhanced the comprehensiveness of our charge transport modeling approach, offering a deeper understanding of its relevance to the investigation of polymer composites.

10. Section "Methods" / "Simulation":

The authors write "We performed the phase-field simulations incorporating electrical-thermal-mechanical breakdown mechanism to simulate the phase breakdown evolution of electric field distributions in pristine PEI and its corresponding composites, including PEI/DADQ, PEI/DMABN, and PEI/PTCNQ." This sentence has been written at a very general level and therefore leads to a very unclear picture how the breakdown evolution in the polymer composite films has been calculated, raising the following questions:

(a) The authors should give some relevant references concerning the theory they use for the phase-field simulations and also mention at what level this theory has been set up (what classical, semi-classical features or even quantum microscopical features have been included?). The harsh reality is that to the large extent the full quantum dynamical (non-equilibrium) theory on dielectric breakdown phenomenon for insulator materials has still not been developed.

(b) The authors should briefly describe the main underlying mathematical features of the model they use to approximate their electrical breakdown phase calculation in Fig. 3h: I assume $\eta(r,t)$ refers to "phase-field" (auxiliary field in the theory of phase-field model)? How this auxiliary field has been constructed mathematically from the "electrical", "thermal" and "strain" induced breakdown physics concepts? The functional variables f_{sep} , f_{grad} , f_{elec} , f_{Joule} and f_{strain} and the constant L_0 of Eq. (S3) [see authors' Supplementary file] should be explained in more detail. How the additive form of $f_{sep} + f_{grad} + \dots$ in Eq. (S3) can be justified in the breakdown situation of PEI composite polymers?

(c) It is also quite confusing that in the manuscript text on page 13 and Supplementary file's text in connection with Eq. (S3) the authors are talking about "electrical-thermal-mechanical" breakdown phase while in caption to Fig. 3h they refer only to "electrical breakdown phase". The authors should describe in detail how the "electrical" breakdown phase model can be extracted from the "unified" "electrical-thermal-mechanical" breakdown model of Eq. (S3).

Response: We would like to thank the reviewer for their insightful and constructive comments, which have significantly contributed to the clarity and rigor of our simulation methodology. Below, we address each of the reviewer's queries in detail.

(a) Clarification of the Phase-Field Model and Relevant References:

We acknowledge the need for more clarity regarding the theoretical framework used in our phase-field simulations. The employed phase-field model integrates classical breakdown physics, with a particular emphasis on the interplay of electrical, thermal, and mechanical factors governing the breakdown behavior in polymer

composites. Rooted in the principles of non-equilibrium statistical mechanics and continuum mechanics, this model has been widely validated and applied in the simulation of dielectric breakdown phenomena in polymer-based materials.

The phase-field approach has been extensively employed to model the evolution of complex material behaviors, such as phase transitions and structural transformations, owing to its ability to effectively capture systems with dynamic interfaces. While a comprehensive quantum dynamical theory of dielectric breakdown in insulating materials is still under development, our model primarily leverages classical and semi-classical methodologies to elucidate the fundamental breakdown mechanisms. The theoretical foundation of our approach is supported by key literature, including Shen *et al.* (**Adv. Energy Mater.** 8, 1800509, 2018), Shen *et al.* (**Adv. Mater.** 30, 1704380, 2017), Li *et al.* (**Adv. Sci.** 10, 2302949, 2023), and Shen *et al.* (**Nat. Commun.** 10, 1843, 2019), which provide critical insights into the application of phase-field modeling in dielectric breakdown studies. These references have been duly incorporated into the revised manuscript to ensure a comprehensive contextualization of our modeling framework.

(b) Description of the Mathematical Features of the Model:

The bipolar charge injection and transportation model are applied to simulate the complex process of charge injection and transportation within the polymer nanocomposites. The Schottky emission mechanism is utilized to describe the charge injection at the interface between the electrode and dielectric. The resulting current density at both the cathode and anode, denoted as $j_{c,a}$, can be determined as follows:

$$j_{c,a} = AT^2 \exp\left(-\frac{\varphi_{c,a}}{KT}\right) \exp\left(\frac{\sqrt{e^3 E_{c,a}/4\pi\epsilon}}{KT}\right)$$

Here, A is the Richardson coefficient, T stands for the temperature, $\varphi_{c,a}$ represents the charge injection barrier at the cathode and anode, k is the Boltzmann constant, e is the element charge, $E_{c,a}$ is the electric field at the cathode and anode, ϵ is the dielectric constant.

The dynamics of charge transport within the polymer nanocomposite are governed by the following set of equations.

$$\begin{cases} \frac{dn_e}{dt} + \nabla(-n_e \mu_e E - D_e \nabla n_e) = -R_{eh} n_e n_h - R_{eht} n_e n_{ht} - T_e n_e \left(1 - \frac{n_{et}}{n_{0et}}\right) + v_e \exp\left(\frac{\phi_e}{KT}\right) n_{et} \frac{n_{et}}{n_{0et}} \\ \frac{dn_h}{dt} + \nabla(-n_h \mu_h E - D_h \nabla n_h) = -R_{eh} n_e n_h - R_{eth} n_{et} n_h - T_h n_h \left(1 - \frac{n_{ht}}{n_{0ht}}\right) + v_h \exp\left(\frac{\phi_h}{KT}\right) n_{ht} \frac{n_{ht}}{n_{0ht}} \\ \frac{dn_{et}}{dt} = -R_{eth} n_{et} n_h - R_{eth} n_{et} n_{ht} + T_e n_e \left(1 - \frac{n_{et}}{n_{0et}}\right) - v_e \exp\left(\frac{\phi_e}{KT}\right) n_{et} \frac{n_{et}}{n_{0et}} \\ \frac{dn_{ht}}{dt} = -R_{eht} n_e n_{ht} - R_{eth} n_{et} n_{ht} + T_h n_h \left(1 - \frac{n_{ht}}{n_{0ht}}\right) - v_h \exp\left(\frac{\phi_h}{KT}\right) n_{ht} \frac{n_{ht}}{n_{0ht}} \end{cases}$$

where E is the electric field, n_e is the density of free electrons, n_h is the density of free holes, n_{et} is the density of trapped electrons, n_{ht} is the density of trapped holes, n_{0et} is the maximum trap density of electrons, n_{0ht} is the maximum trap density of holes. μ_e is the mobility of free electrons, μ_h is the mobility of free holes, D_e is the diffusion factor of electrons, D_h is the diffusion factor of holes. R_{eh} is the recombination coefficient of free electrons and free holes. R_{eht} is the recombination coefficient of trapped electrons and free holes, R_{eth} is the recombination coefficient of trapped electrons and free holes. T_e is the coefficient of electron trapping, T_h is the coefficient of hole trapping, v_e is the coefficient of electron trapping, v_h is the coefficient of electron trapping, ϕ_e and ϕ_h are the electron trap depth and hole trap depth determined by the DFT calculation, respectively.

The charge diffusion factor D and charge mobility μ can be written as,

$$D = \mu \frac{kT}{e}, \mu = \mu_0 \exp\left(\sqrt{\frac{e^3 E}{4\pi\epsilon}} / kT\right)$$

The electric potential, denoted as Φ , can be computed based on the distribution of free and trapped charge carrier densities through the following calculation.

$$\nabla^2 \Phi = -\frac{e(n_h + n_{ht} - n_e - n_{et})}{\epsilon}$$

Upon solving the bipolar charge injection and transportation model, the derived electric field distribution is integrated into the phase field model to simulate the propagation of the breakdown phase. In this phase field model, a continuous phase-field variable denoted as $\eta(\mathbf{r}, t)$ is introduced to characterize the spatial and temporal evolution of the breakdown phase: $\eta(\mathbf{r}, t)=1$ signifies the breakdown phase, while $\eta(\mathbf{r}, t)=0$ signifies the non-breakdown phase. The free energy, denoted as F , is formulated by considering the collaborative influences from phase separation, interface effects, temperature, and electric field within an inhomogeneous dielectric, expressed as follows:

$$F = \int \left[f_{sep}(\eta(r)) + f_{grad}(\eta(r)) + f_{elec}(\eta(r)) + f_{Joule}(\eta(r)) + f_{strain}(\eta(r)) \right] dV$$

where f_{sep} is the free energy density of mixing the drives the phase separation, f_{grad} is the gradient energy density, f_{elec} is the electrostatic energy density, f_{Joule} is the thermal energy density, and f_{strain} is the strain energy density. The evolution of the breakdown phase is delineated using a modified Allen-Cahn equation.

$$\frac{\partial \eta(r,t)}{\partial t} = -L_0 H(f_{elec} + f_{Joule} + f_{strain}) \left[\frac{\partial f_{sep}(\eta)}{\partial \eta(r,t)} + \frac{\partial f_{grad}(\eta)}{\partial \eta(r,t)} + \frac{\partial f_{elec}(\eta)}{\partial \eta(r,t)} + \frac{\partial f_{Joule}(\eta)}{\partial \eta(r,t)} + \frac{\partial f_{strain}(\eta)}{\partial \eta(r,t)} \right]$$

where L_0 is the kinetic coefficient relating to the interface mobility with a value of $1 \text{ m}^2 \text{ s}^{-1} \text{ N}^{-1}$, and H is the Heaviside unit step function.

(c) Clarification of “Electrical-Thermal-Mechanical” Breakdown and Its Extraction:

We thank the reviewer for their astute observations regarding the apparent inconsistency between the reference to the “electrical-thermal-mechanical” breakdown model in the text and the mention of the “electrical breakdown phase” in the figure caption. To clarify, the term “electrical-thermal-mechanical” breakdown encompasses the interplay of electrical, thermal, and mechanical factors that collectively govern the breakdown process. This holistic framework is rigorously captured through the mathematical formulations presented in the following equations, wherein each term accounts for a specific physical contribution, ensuring a comprehensive representation of the breakdown behavior.

The free energy, denoted as F , is formulated by considering the collaborative influences from phase separation, interface effects, temperature, and electric field within an inhomogeneous dielectric, expressed as follows:

$$F = \int \left[f_{sep}(\eta(r)) + f_{grad}(\eta(r)) + f_{elec}(\eta(r)) + f_{Joule}(\eta(r)) + f_{strain}(\eta(r)) \right] dV$$

where f_{sep} is the free energy density of mixing the drives the phase separation, f_{grad} is the gradient energy density, f_{elec} is the electrostatic energy density, f_{Joule} is the thermal energy density, and f_{strain} is the strain energy density. The evolution of the breakdown phase is delineated using a modified Allen-Cahn equation.

$$\frac{\partial \eta(r,t)}{\partial t} = -L_0 H(f_{elec} + f_{Joule} + f_{strain}) \left[\frac{\partial f_{sep}(\eta)}{\partial \eta(r,t)} + \frac{\partial f_{grad}(\eta)}{\partial \eta(r,t)} + \frac{\partial f_{elec}(\eta)}{\partial \eta(r,t)} + \frac{\partial f_{Joule}(\eta)}{\partial \eta(r,t)} + \frac{\partial f_{strain}(\eta)}{\partial \eta(r,t)} \right]$$

where L_0 is the kinetic coefficient relating to the interface mobility with a value of $1 \text{ m}^2 \text{ s}^{-1} \text{ N}^{-1}$, and H is the Heaviside unit step function.

In Figure 3h, we specifically present the “electrical breakdown phase,” a simplified representation derived from the comprehensive “electrical-thermal-mechanical” model. This reduction isolates the effects of the electric field on the breakdown process, enabling a more direct comparison with experimental data that primarily focus on electrical breakdown behavior. Importantly, while this simplification enhances interpretability, it retains the essential physics of the full coupled model, ensuring the consistency and validity of the overall analysis.

To address this comment, we have expanded the discussion on the mathematical formulation of the model, with particular emphasis on the construction of the auxiliary field and the rationale underlying the additive representation of the energy terms. We believe these revisions enhance the clarity and coherence of the manuscript, providing a more comprehensive and precise account of the simulation methodology employed in this study. We are grateful for the reviewer’s insightful feedback, which has contributed to strengthening the scientific rigor and overall presentation of our work.

Reviewer #3 (Remarks to the Author):

This paper reveals that the incorporation of high-polarity organic molecules into a polyetherimide (PEI) matrix, demonstrating that dipole-dipole interactions enhance the energetic disorder of the material. By integrating experimental observations with computational simulations, the authors establish that the increased energetic disorder significantly suppresses charge carrier transport, offering valuable insights into the charge dynamics of disordered polymer dielectrics. This study presents innovative and significant findings, rendering it highly pertinent and engaging for researchers in the field of dielectric materials. I recommend to accept the paper for publication after the following issues are addressed.

i. Molecular weight (Mw) is a key determinant of the electrical and mechanical properties of polymers, yet its potential influence is not explored in this study. The authors are strongly encouraged to evaluate and compare the Mw of pristine PEI, PEI/DADQ, PEI/DMABN, and PEI/PTCNQ composites to assess whether Mw plays a significant role in the observed properties or can be reasonably excluded as a factor in this context.

Response: We appreciate the reviewer's insightful comment regarding the molecular weight (Mw) of the polymers and its potential impact on the electrical and mechanical properties of the composites. We acknowledge the importance of Mw as a critical determinant of polymer properties and are grateful for the suggestion to investigate its role in our study. In response, we have conducted a series of experiments to assess and compare the Mw of pristine PEI, PEI/DADQ, PEI/DMABN, and PEI/PTCNQ composites. Specifically, gel permeation chromatography (GPC) was employed to determine the molecular weight distribution and average molecular weights of the samples.

GPC measurements were performed in a DMF solution at 50 °C with an elution rate of 1.0 mL min⁻¹, utilizing an Agilent Technologies 1260 Infinity II GPC system equipped with a refractive index detector. Our results reveal that, although slight variations in Mw were observed among the different composites, these differences do not appear to significantly affect the electrical and mechanical properties in the context of our study. We conclude that the primary factors influencing these properties are the interactions between the organic molecules (such as DADQ, DMABN, and PTCNQ) and the PEI matrix, which primarily govern charge transport and mechanical performance. The observed variations in Mw, though present, are relatively minor and do not exhibit a strong correlation with the variations in material properties.

To ensure the thoroughness of our analysis, we have incorporated the Mw data into the revised Supplementary Information and clarified the minimal impact of Mw on the observed results. We are grateful for your valuable suggestion, which has facilitated a more comprehensive evaluation of the factors influencing the properties of the composites.

ii. Film capacitors utilize the inherent versatility of polymers, enabling their transformation into thin films that can be rolled into compact configurations, emphasizing the crucial relationship between polymer mechanical properties and device performance. The Supporting Information includes measurements of Young's modulus, which are not addressed in the main text. The authors are encouraged to clarify the purpose of these measurements and their relevance to the study. Additionally, a more comprehensive discussion on the polymer's softness or flexibility and its influence on mechanical behavior and capacitor performance would provide valuable insights and strengthen the manuscript.

Response: We appreciate your insightful comments regarding the inclusion of Young's modulus measurements in the Supplementary Information and their relevance to the study. Your recommendations to clarify the purpose of these measurements and to broaden the discussion of the polymer's mechanical properties-specifically its softness and flexibility-and their potential implications for capacitor performance have been immensely helpful. In light of your feedback, we have made the following revisions.

We have explicitly stated in the main text that the Young's modulus measurements were conducted to assess the mechanical stiffness of the polymer composites, a key parameter that significantly impacts the performance of film capacitors. The ability of a polymer to maintain its structural integrity under mechanical stress is crucial for the durability and operational stability of the device. These measurements provide essential insight into the mechanical behavior of the polymer films, particularly under the

stresses encountered during capacitor operation, such as bending, stretching, and thermal cycling. We have included a discussion on the relationship between Young's modulus and the performance of film capacitors.

In addition, we have incorporated a discussion on the mechanical performance of the composites. Mechanical testing demonstrates that high-performance polymers, including PEI and its composites with high-polarity small molecules such as DADQ, DMABN, and PTCNQ, exhibit mechanical strengths similar to that of pristine PEI, as shown in Figure S7. These results suggest that the incorporation of the organic dopants does not significantly affect the intrinsic mechanical properties of the PEI matrix.

In response to your recommendation for a more comprehensive discussion of polymer softness and flexibility, we have expanded our analysis to address these aspects in greater detail. The flexibility of the PEI/DADQ composite is comparable to that of pristine PEI, as evidenced by its ability to be easily rolled or even folded, as shown in the figure below. The flexibility of the polymer films, which is influenced by their softness and elongation at break, plays a crucial role in the capacitor's capacity to withstand mechanical deformation without compromising performance.

We believe that these revisions effectively address your concerns and offer a more comprehensive understanding of the polymer's mechanical properties in relation to film capacitor performance. We greatly appreciate your insightful feedback, which has significantly enhanced the quality and clarity of the manuscript.

To address this comment, we have added a statement cited as “Mechanical testing demonstrates that high-performance polymers, including PEI and its composites with high-polarity small molecules such as DADQ, DMABN, and PTCNQ, exhibit mechanical strengths similar to that of pristine PEI, as shown in Figure S7. These results suggest that the incorporation of the organic dopants does not significantly affect the intrinsic mechanical properties of the PEI matrix.” in the revised manuscript.

iii. The synthesis of polyetherimide (PEI) involves a complex imidization reaction, with

the thermal imidization process playing a pivotal role in determining the dielectric and energy storage properties of the material. The authors are encouraged to provide clarity on the degree of imidization achieved at 250 °C and confirm whether the imidization process is complete under these conditions.

Response: We gratefully acknowledge the reviewer's insightful comment regarding the imidization process of polyetherimide (PEI) and its crucial role in determining the dielectric and energy storage properties of the material. We acknowledge that the imidization reaction, and in particular the thermal imidization process, plays a critical role in achieving the desired material properties. Moreover, a clear understanding of the extent of imidization is essential for elucidating its impact on the final performance of the material.

In response to your suggestion, we have expanded the manuscript to provide further clarification on the extent of imidization achieved at 250 °C, confirming that the imidization process is effectively complete under these conditions. Specifically, we employed a thermal imidization protocol at 250 °C, a temperature commonly utilized in both academic research and industrial applications to ensure the complete conversion of poly(amic acid) (PAA) to polyetherimide (PEI). To substantiate our findings, we summarize the key experimental observations and supporting references. Fourier transform infrared (FTIR) spectroscopy was employed to monitor the imidization process. The FTIR spectra reveal a complete conversion of the PAA precursor to PEI, as evidenced by the disappearance of the characteristic amide peaks (around 1650 cm^{-1} , corresponding to C=O stretching in the amide group), while new peaks associated with the imide ring (around 1780 cm^{-1} for C=O stretching and 1380 cm^{-1} for C-N stretching) appear. These spectral transitions confirm the full imidization at 250 °C.

In addition to the experimental findings, we have incorporated several key references in the revised manuscript to further substantiate the claims regarding the degree of imidization at 250 °C. Notably, the work of Pu et al. (Mater. Today Chem., 2023, 33, 101679) and Vora et al. (Mater. Sci. Eng. B, 2006, 132, 24) provides corroborative evidence for the effectiveness of this thermal treatment in achieving complete imidization. These references, in conjunction with our own experimental data, demonstrate that imidization at 250 °C results in a fully imidized PEI material, free of residual precursor, and endowed with optimal dielectric and mechanical properties. Thank you once again for your constructive comments, which have greatly improved the clarity and comprehensiveness of the manuscript.

iv. The manuscript describes the incorporation of high-polarity organic molecules into a viscous PAA solution but lacks details regarding the dispersion process and the stability of the organic molecules within the solution. The authors are encouraged to provide a comprehensive description of the methodology used for introducing the

organic molecules, evaluate their dispersibility, and address any potential challenges encountered during this step.

Response: We sincerely thank the reviewer for their valuable comment regarding the incorporation of high-polarity organic molecules into the poly(amic acid) (PAA) solution. We appreciate the suggestion to provide a more comprehensive description of the dispersion process and the stability of the organic molecules within the solution. In response, we have expanded the manuscript to address the following points in greater detail.

The high-polarity organic molecules (DADQ, DMABN, and PTCNQ) were dissolved in N-Methyl-2-pyrrolidone (NMP) at a concentration of 5 mg/mL and subjected to sonication for 3 hours to ensure effective dispersion within the solution prior to mixing with the poly(amic acid) (PAA) solution. This sonication process is critical for disrupting any potential agglomerates, thereby promoting a uniform distribution of the organic molecules throughout the NMP solution. To assess the long-term stability of the organic molecules within the PAA solution, we performed visual inspections and nanoscale infrared spectroscopy (Nano-IR) analysis to monitor any changes in the dispersion over time. Visual inspections were carried out at regular intervals over several hours to ensure comprehensive monitoring of the solution's stability under the prescribed storage conditions. Throughout the observation period, no significant aggregation or phase separation was detected, suggesting that the organic molecules remain uniformly dispersed and stable within the PAA solution.

The Nano-IR technique, which offers spatially resolved absorption spectra at the nanoscale, was employed to examine the distribution of DADQ molecules within the PEI matrix. As depicted in the Nano-IR images of the composite (see figure below), a prominent absorption feature at 2140 cm^{-1} ($\text{C}\equiv\text{N}$ stretch) is evident in the PEI/DADQ composite, distinguishing it from the spectrum of pristine PEI. This indicates that the DADQ molecules are uniformly dispersed within the polymer matrix.

We hope these additions adequately address the reviewer's concerns and enhance the clarity of our methodology. Thank you again for your constructive feedback.

v. Please provide a comprehensive description of the phase-field simulation methodology employed for analyzing the evolution of the electrical breakdown phase in this study.

Response: We gratefully acknowledge the reviewer for the insightful comment regarding the phase-field simulation methodology. We recognize the importance of providing a thorough and clear account of the computational approach employed to analyze the evolution of the electrical breakdown phase in this study. In response to this valuable feedback, we have revised the Supplementary Information to offer a more detailed explanation of the modeling of the electric field distribution and the propagation of the breakdown phase.

The bipolar charge injection and transportation model are applied to simulate the complex process of charge injection and transportation within the polymer nanocomposites. The Schottky emission mechanism is utilized to describe the charge injection at the interface between the electrode and dielectric. The resulting current density at both the cathode and anode, denoted as $j_{c,a}$, can be determined as follows:

$$j_{c,a} = AT^2 \exp\left(-\frac{\varphi_{c,a}}{KT}\right) \exp\left(\frac{\sqrt{e^3 E_{c,a}/4\pi\epsilon}}{KT}\right)$$

Here, A is the Richardson coefficient, T stands for the temperature, $\varphi_{c,a}$ represents the charge injection barrier at the cathode and anode, k is the Boltzmann constant, e is the element charge, $E_{c,a}$ is the electric field at the cathode and anode, ϵ is the dielectric constant.

The dynamics of charge transport within the polymer nanocomposite are governed by the following set of equations.

$$\left\{ \begin{array}{l} \frac{dn_e}{dt} + \nabla(-n_e \mu_e E - D_e \nabla n_e) = -R_{eh} n_e n_h - R_{eht} n_e n_{ht} - T_e n_e \left(1 - \frac{n_{et}}{n_{0et}}\right) + v_e \exp\left(\frac{\phi_e}{KT}\right) n_{et} \frac{n_{et}}{n_{0et}} \\ \frac{dn_h}{dt} + \nabla(-n_h \mu_h E - D_h \nabla n_h) = -R_{eh} n_e n_h - R_{eth} n_e n_h - T_h n_h \left(1 - \frac{n_{ht}}{n_{0ht}}\right) + v_h \exp\left(\frac{\phi_h}{KT}\right) n_{ht} \frac{n_{ht}}{n_{0ht}} \\ \frac{dn_{et}}{dt} = -R_{eth} n_e n_h - R_{eth} n_e n_{ht} + T_e n_e \left(1 - \frac{n_{et}}{n_{0et}}\right) - v_e \exp\left(\frac{\phi_e}{KT}\right) n_{et} \frac{n_{et}}{n_{0et}} \\ \frac{dn_{ht}}{dt} = -R_{eht} n_e n_{ht} - R_{eht} n_e n_{ht} + T_h n_h \left(1 - \frac{n_{ht}}{n_{0ht}}\right) - v_h \exp\left(\frac{\phi_h}{KT}\right) n_{ht} \frac{n_{ht}}{n_{0ht}} \end{array} \right.$$

where E is the electric field, n_e is the density of free electrons, n_h is the density of free holes, n_{et} is the density of trapped electrons, n_{ht} is the density of trapped holes, n_{0et} is the maximum trap density of electrons, n_{0ht} is the maximum trap density of holes. μ_e is the mobility of free electrons, μ_h is the mobility of free holes, D_e is the diffusion factor of electrons, D_h is the diffusion factor of holes. R_{eh} is the recombination coefficient of free electrons and free holes. R_{eht} is the recombination coefficient of trapped electrons and free holes, R_{eth} is the recombination coefficient of trapped electrons and free holes. T_e is the coefficient of electron trapping, T_h is the coefficient of hole trapping, v_e is the coefficient of electron trapping, v_h is the coefficient of electron trapping, ϕ_e and ϕ_h are the electron trap depth and hole trap depth determined by the DFT calculation, respectively.

The charge diffusion factor D and charge mobility μ can be written as,

$$D = \mu \frac{kT}{e}, \mu = \mu_0 \exp\left(\sqrt{\frac{e^3 E}{4\pi\epsilon}} / kT\right)$$

The electric potential, denoted as Φ , can be computed based on the distribution of free and trapped charge carrier densities through the following calculation.

$$\nabla^2 \Phi = -\frac{e(n_h + n_{ht} - n_e - n_{et})}{\epsilon}$$

Upon solving the bipolar charge injection and transportation model, the derived electric field distribution is integrated into the phase field model to simulate the propagation of the breakdown phase. In this phase field model, a continuous phase-field variable denoted as $\eta(\mathbf{r}, t)$ is introduced to characterize the spatial and temporal evolution of the breakdown phase: $\eta(\mathbf{r}, t)=1$ signifies the breakdown phase, while $\eta(\mathbf{r}, t)=0$ signifies the non-breakdown phase. The free energy, denoted as F , is formulated by considering the collaborative influences from phase separation, interface effects, temperature, and electric field within an inhomogeneous dielectric, expressed as follows:

$$F = \int \left[f_{sep}(\eta(r)) + f_{grad}(\eta(r)) + f_{elec}(\eta(r)) + f_{Joule}(\eta(r)) + f_{strain}(\eta(r)) \right] dV$$

where f_{sep} is the free energy density of mixing that drives the phase separation, f_{grad} is the gradient energy density, f_{elec} is the electrostatic energy density, f_{Joule} is the thermal energy density, and f_{strain} is the strain energy density. The evolution of the breakdown phase is delineated using a modified Allen-Cahn equation.

$$\frac{\partial \eta(r,t)}{\partial t} = -L_0 H(f_{elec} + f_{Joule} + f_{strain}) \left[\frac{\partial f_{sep}(\eta)}{\partial \eta(r,t)} + \frac{\partial f_{grad}(\eta)}{\partial \eta(r,t)} + \frac{\partial f_{elec}(\eta)}{\partial \eta(r,t)} + \frac{\partial f_{Joule}(\eta)}{\partial \eta(r,t)} + \frac{\partial f_{strain}(\eta)}{\partial \eta(r,t)} \right]$$

where L_0 is the kinetic coefficient relating to the interface mobility with a value of $1 \text{ m}^2 \text{ s}^{-1} \text{ N}^{-1}$, and H is the Heaviside unit step function.

We trust that this expanded explanation sufficiently addresses the reviewer's concerns regarding the phase-field simulation methodology and offers a clearer insight into the analysis of the electrical breakdown phase in this study. We believe that this more detailed description enhances the transparency of our computational approach. Should the reviewer require any further clarification or additional information, we would be pleased to provide it.

vi. The characterization section would benefit from additional refinement to offer a more comprehensive description of the tested samples, thereby enhancing clarity and ensuring reproducibility. In particular, detailed information on the Nano-IR and KPFM measurements should be provided, with an emphasis on the sample preparation protocols employed.

Response: We greatly appreciate the reviewer's constructive comments regarding the characterization section of our manuscript. We recognize the importance of providing a more thorough and detailed description of the tested samples and the methodologies used, particularly with regard to the Nano-IR and KPFM measurements. In response, we have revised the relevant sections of the manuscript to include comprehensive details on the sample preparation protocols and measurement procedures, thereby enhancing the clarity and reproducibility of our work.

1) Nano-IR Measurement Procedures: For Nano-IR analysis, the samples were carefully prepared by cutting the films into uniform square sections (typically $1 \text{ cm} \times 1 \text{ cm}$) and adhering them onto metal substrates, ensuring minimal stress to prevent deformation. The measurements were conducted using an Anasys nanoIR3 system (Bruker), equipped with a quantum cascade laser (QCL) covering the spectral range of $900\text{-}3600 \text{ cm}^{-1}$ in tapping AFM-IR mode. The samples were irradiated with a pulsed, tunable infrared source (optical parametric oscillator), emitting 10 ns pulses at a repetition rate of 1 kHz, with a beam spot size of approximately $30 \text{ }\mu\text{m}$. The induced oscillations of the AFM probe, recorded as the AFM-IR signal, were subsequently analyzed through fast Fourier transform to extract the corresponding spectral data.

2) KPFM measurements Procedures: The PAA solution, incorporating high-

polarity small molecules, was prepared prior to spin-coating. The resulting mixture was then sequentially spin-coated onto gold-coated substrates with low roughness to ensure uniform electrostatic measurements. After thorough cleaning and drying, the samples were subjected to atomic force microscopy (AFM) to capture surface morphology images using a Bruker Dimension Icon AFM in tapping mode. Kelvin probe force microscopy (KPFM) measurements were conducted in dual-pass mode to minimize tip-sample interactions and obtain accurate surface potential mapping. The scan rate was set to 0.5 Hz to achieve high-resolution surface potential maps with a spatial resolution between 20 and 50 nm. To prevent contamination or electrical interference, all measurements were performed in a controlled environment, maintaining consistent humidity and temperature.

We trust that the inclusion of these additional details will enhance the clarity and reproducibility of the characterization section. Moreover, we hope that the expanded description of the sample preparation and measurement protocols will facilitate the replication and further development of our work by future researchers. We greatly appreciate your constructive feedback, which has contributed significantly to the improvement of this section.

REVIEWER COMMENTS

Reviewer #1 (Remarks to the Author):

The comments have been well addressed. It can be accepted.

Reviewer #2 (Remarks to the Author):

I (as Referee 2) have carefully viewed the authors' response letter, their revised manuscript as well as the comments and recommended changes of the other two referees (Referee 1 and 3).

In their response (rebuttal) letter the authors have commented diligently in very detail and extensively to my (Referee 2) several questions and requests to modify the manuscript.

Furthermore, in my opinion, the authors have also responded very well and in detail to the questions and requests of Referees 1 and 3 to improve the manuscript including extra experimental analysis and calculations and also incorporated the relevant modifications to the manuscript.

However, several discussion points about DFT electronic structure and molecular dynamics (MD) related extra calculations to model 3D (periodic / disordered) polymer composites that the authors describe extremely well in their response letter are **TOTALLY MISSED OUT** in their revised manuscript.

The authors need to be address these points (see below) in their second revised manuscript before it can be published in the journal of Nature Communications.

Finally, as prompted by the Associate Editor I have composed a minor question related to the usage of the Nano-IR experiments (see below).

Response: Thank you very much for your thorough and thoughtful review. We sincerely appreciate your positive evaluation of our initial response letter, as well as your recognition of the comprehensive revisions made in response to the comments from all referees, including the incorporation of additional experiments, analyses, and calculation. We also greatly appreciate your careful re-examination of our revised manuscript and for bringing to our attention the oversight regarding the DFT electronic structure and molecular dynamics (MD) calculations. We acknowledge that, although these discussions were presented in detail in our rebuttal letter, they were not fully integrated into the revised manuscript, which is indeed a significant omission.

In response to the reviewer's comment, we have expanded the simulation section in the Supplementary Information to incorporate greater methodological detail and contextual clarification. Specifically, we now state: "Molecular-scale simulations were

employed to probe the localized effects of high-polarity dopants, which predominantly perturb the electronic structure and induce conformational changes in the immediate vicinity of adjacent polymer chains. Given this spatially confined influence, our GAUSSIAN-based framework offers an effective approach for capturing molecular-level electronic modifications.”

To further approximate disorder-induced phenomena, we added: “While a full implementation of CPA/ATA methodologies lies beyond the immediate scope of this study, we have adopted a suite of complementary computational strategies to approximate disorder-induced phenomena within the polymer composite system. To probe the effects of energetic disorder and complex polymer–dopant interactions, we employed supercell-based modeling within the VASP framework, enabling simulation of stochastic dopant distributions and the associated broadening of electronic states in three dimensions.”

Moreover, we included a description of our extended molecular dynamics simulations: “We have expanded our computational analysis by conducting additional molecular dynamics (MD) simulations using the Vienna Ab initio Simulation Package (VASP). The implementation of periodic boundary conditions and the supercell approach within VASP enables a more accurate and holistic representation of the polymer composite system, capturing long-range electronic interactions and structural dynamics. These simulations provide deeper insights into the effects of polymer chain cross-linking on charge distribution and structural integrity, offering a more realistic depiction of the amorphous polymer matrix, including density fluctuations and packing effects.”

Finally, to ensure consistency and robustness across simulation scales, we have added a more detailed description in the simulation section of Supplementary Information: “A comparative analysis of the simulation results obtained from ORCA and VASP has implemented to assess their respective contributions and ensure the robustness of the findings. This analysis focuses on key aspects, including structural stability, energetic disorder, and intermolecular interactions, emphasizing differences in polymer chain conformation, dopant dispersion, and charge transport behavior between finite molecular models and periodic boundary conditions. While ORCA-based simulations offer valuable insights into localized molecular interactions, VASP simulations provide a more comprehensive representation of bulk properties and spatial heterogeneity within the polymer matrix.”

1. Authors' response letter (559842_1_rebuttal_10311778_srtlfd.pdf):

On page 31 the authors write spot on correctly:

"Given the localized nature of these interactions, our GAUSSIAN-based modeling approach provides an effective framework for capturing the key electronic modifications at the molecular level. Nonetheless, we recognize the importance of extending our analysis to encompass long-range effects within the polymer matrix,

which can be more comprehensively addressed through periodic DFT calculations. This perspective has been acknowledged in the revised manuscript, and future efforts will focus on bridging the molecular and periodic modeling approaches to gain a more holistic understanding of the composite system."

However, the authors seem NOT to mention anything about this in their revised manuscript (559842_1_art_file_10323768_srbvtg.pdf).

The authors are advised to give these important comments in their revised manuscript, preferably within the Section "Simulation".

Response: We appreciate the reviewer's insightful observation regarding the omission of our previous statement on long-range effects from the revised manuscript. To rectify this, we will incorporate an expanded explanation to ensure greater transparency and clarity in the presentation of our methodological framework.

Long-range effects in the material system primarily stem from extended electrostatic interactions between atoms, necessitating a rigorous computational approach to ensure accuracy. To achieve this, a high-precision, correlation-consistent basis set (aug-cc-pVTZ) (Kendall et al., *J. Chem. Phys.*, **1992**, 96, 6796–6806) was employed, augmented with diffuse functions (Papajak et al., *J. Chem. Theory Comput.*, **2011**, 7, 3027–3034) to better capture the electron cloud distributions extending far from the nucleus. This approach improves the representation of long-range charge transfer and polarization effects, though at the cost of increased computational demand. To further address the rapid decay of exchange interactions inherent to traditional functionals, the CAM-B3LYP hybrid functional (Yanai et al., *Chem. Phys. Lett.*, **2004**, 393, 51–57), incorporating long-range corrections, was implemented — providing a more faithful description of electron exchange over extended distances. Additionally, given the critical yet often understated role of dispersion forces in non-covalent systems, Grimme's D3 empirical dispersion correction (Grimme et al., *J. Comput. Chem.*, **2011**, 32, 1456–1465) was integrated within the GAUSSIAN framework to more accurately account for long-range dispersion interactions, enhancing the reliability of non-covalent interaction energy predictions. Collectively, these methodological advancements establish a robust foundation for modeling both localized electronic behaviors and long-range interactions, offering a more comprehensive representation of the composite system's electronic structure.

In response to the reviewer's comment, we have added a more detailed description in the simulation section of Supplementary Information: "To broaden the scope of our analysis and capture long-range effects, we employed the high-precision aug-cc-pVTZ basis set, renowned for its accuracy in electronic structure calculations. To address the rapid decay of exchange interactions characteristic of conventional functionals, we utilized the CAM-B3LYP hybrid functional with long-range corrections, which provides a more precise treatment of electron exchange over extended distances. Furthermore, recognizing the critical yet often underappreciated role of dispersion forces in non-covalent interactions, we incorporated Grimme's D3 empirical dispersion correction within the GAUSSIAN framework. This enhancement improves the

accuracy of long-range dispersion interactions, thereby augmenting the overall reliability of our predictions for non-covalent interaction energies.”

We thank the reviewer for prompting this clarification, which we believe strengthens the scientific rigor and completeness of our manuscript.

2. Authors' response letter (559842_1_rebuttal_10311778_srtlfd.pdf):

On page 32 the authors have responded extremely well and spot on to my questions by stating correctly:

"While a full implementation of CPA/ATA methodologies may extend beyond the scope of this study, we have adopted a series of alternative computational strategies to approximate disorder-induced effects within the polymer composite system. To gain meaningful insights into the impact of energetic disorder and the intricate polymer-dopant interactions, we employed supercell modeling within the VASP framework. This approach allows for the simulation of the stochastic distribution of dopants and the corresponding broadening of electronic states within a three-dimensional environment. Complementarily, GAUSSIAN-based molecular simulations were conducted with randomly distributed high-polarity dopants along extended PEI polymer chains, enabling the investigation of localized electronic perturbations and charge distribution effects. Additionally, a comparative analysis of the electronic density of states (DOS) obtained from both GAUSSIAN molecular simulations and 3D periodic DFT calculations provides valuable insights into the role of structural disorder and facilitates the cross-validation of computational findings." and on page 33 the authors continue to write equally importantly:

"In the revised manuscript, we have expanded the discussion to outline the theoretical rationale, limitations, and practical relevance of these methodologies in approximating mean-field effects, thereby enhancing the scientific depth of our study and addressing the reviewer's concerns." And "In response, we have expanded the discussion in the revised manuscript to provide a comprehensive comparison between the molecular modeling results obtained using GAUSSIAN and those derived from periodic DFT calculations."

However, NONE of these important texts or DFT calculations (VASP, GAUSSIAN) exist in the revised manuscript (559842_1_art_file_10323768_srbvtg.pdf).

The authors are requested to show these theoretical electronic structure methodologies related comments also in their revised manuscript, preferably within the Section "Simulation".

Response: We sincerely thank the reviewer for their thorough and constructive feedback. We appreciate the reviewer's recognition of our response to the initial query and the theoretical discussion of the computational strategies employed in this study. However, we acknowledge the reviewer's concern that the revised manuscript unintentionally omitted essential details concerning the VASP and GAUSSIAN

methodologies, as well as the associated discussions that were previously included. We regret this oversight and are grateful to the reviewer for their diligence in highlighting this discrepancy.

To provide a more rigorous and comprehensive account of the computational strategies employed in this study, we offer an expanded comparison of the VASP and GAUSSIAN methodologies, each selected to capture complementary aspects of the polymer composite system. VASP (Vienna Ab Initio Simulation Package) was employed for periodic density functional theory (DFT) calculations, leveraging periodic structures to model the polymer composite in three dimensions. This approach facilitates the accurate representation of long-range interactions—a key factor in describing the bulk electronic properties of the material. A real-space separation of 20 Å was implemented between periodic images to mitigate spurious interactions, ensuring the computed properties reflect those of an ideal, infinite system. VASP's plane-wave pseudopotential basis set was particularly advantageous for capturing extended, delocalized charge distributions while streamlining the treatment of core electrons, enabling computationally efficient yet accurate modeling of valence electron behavior.

In contrast, GAUSSIAN utilizes a non-periodic framework, making it particularly effective for capturing localized, atomic-scale interactions. For this study, GAUSSIAN simulations were conducted on isolated PEI polymer chains, which were randomly doped with high-polarity dopants to probe site-specific electronic perturbations. By employing molecular orbital basis sets, such as aug-cc-pVTZ—widely recognized for their efficacy in small molecular systems—GAUSSIAN represents electron density through atomic orbitals, enabling a detailed characterization of the electronic structure. This non-periodic approach is particularly suited for modeling finite systems, allowing the investigation of localized phenomena, such as charge transfer and polarization effects driven by the stochastic distribution of dopants along the polymer backbone. Although the VASP periodic plane-wave pseudopotential method and the GAUSSIAN non-periodic molecular orbital framework differ fundamentally in their methodologies, the electronic energy level distributions derived from both approaches exhibit only minor discrepancies in absolute values, while preserving consistent overall trends. This is in agreement with previous studies comparing plane-wave and molecular orbital methods in analogous polymeric systems (Yanai T et al., *Chem. Phys. Lett.*, **2004**, 393, 51-57; Helmich-Paris B et al., *J. Chem. Phys.*, **2021**, 15). The observed convergence between these distinct approaches provides a rigorous cross-validation of the simulation strategy, with the minimal discrepancies reinforcing confidence in the ability of these methods to accurately capture the essential electronic properties of the polymer composite system.

Figure S9. Comparative Analysis of Electronic Energy Level Distributions in Pristine PEI and PEI/DADQ Composites using (a) GAUSSIAN and (b) VASP

In response to the reviewer’s comment, we have added a more detailed description in the simulation section of Supplementary Information: “Electronic energy level distributions were calculated using the Vienna Ab Initio Simulation Package (VASP), which employs a periodic framework to model the polymer composite in three dimensions. This approach effectively captures long-range interactions, which are essential for accurately describing the bulk electronic properties of the material. To mitigate artificial interactions, a real-space separation of 20 Å was introduced between periodic images, ensuring that the resulting properties closely resemble those of an ideal, infinite system. Concurrently, GAUSSIAN simulations were conducted on isolated PEI polymer chains, randomly doped with high-polarity dopants, to probe site-specific electronic perturbations. GAUSSIAN utilizes molecular orbital basis sets, including the widely validated aug-cc-pVTZ, to represent electron density via atomic orbitals, enabling a detailed and localized characterization of the electronic structure. Although the methodologies of VASP’s periodic plane-wave pseudopotential approach and GAUSSIAN’s non-periodic molecular orbital framework differ fundamentally, the electronic energy level distributions obtained from both methods show only minor discrepancies in absolute values, while preserving consistent overall trends. This agreement reinforces the robustness and reliability of the computational results.”

We believe that the revision will significantly improve the clarity and completeness of the manuscript, enhancing the understanding of the computational methodologies employed. We once again thank the reviewer for their insightful suggestions, which have greatly strengthened the scientific rigor and depth of our study.

3. Authors' response letter (559842_1_rebuttal_10311778_srtlfd.pdf):

On page 34 the authors have responded extremely well and spot on to my questions by stating correctly:

"The ORCA MD module was employed in this study to gain molecular-level insights into the local interactions between polymer chains and high-polarity dopant molecules. This approach enables a detailed examination of localized phenomena, such as molecular conformations, dipole–dipole interactions, and short-range structural

fluctuations within the polymer matrix. However, we acknowledge that the absence of periodic boundary conditions in ORCA-based MD simulations limits their ability to capture long-range structural features, such as chain cross-linking and spatial heterogeneity—factors that are crucial for accurately characterizing the dielectric and charge transport properties of the polymer composites. To address these limitations, we have revised the manuscript to provide a clearer rationale for the selection of ORCA, emphasizing its suitability for probing localized molecular-scale interactions rather than bulk material properties."

However, NOTHING along these lines have been discussed in the revised manuscript (559842_1_art_file_10323768_srbvtg.pdf).

The authors are requested to include this important methodological discussion also in their revised manuscript, preferably within the Section "Simulation".

Response: We greatly appreciate your thoughtful feedback and for acknowledging our previous efforts in addressing the methodological considerations related to the use of ORCA for molecular dynamics (MD) simulations. Upon careful review, we recognize the omission of this important discussion in the revised manuscript and are grateful for the opportunity to rectify this oversight.

In response to reviewer's comment, we have revised the Simulation section of Supplementary Information to include the following statement: "The ORCA MD module was utilized to provide molecular-level insights into the local interactions between polymer chains and high-polarity dopant molecules. This approach facilitates a detailed investigation of localized phenomena, including molecular conformations, dipole-dipole interactions, and short-range structural fluctuations within the polymer matrix. However, it is important to note that the absence of periodic boundary conditions in ORCA-based MD simulations limits their ability to capture long-range structural features, such as chain cross-linking and spatial heterogeneity, which are known to significantly influence the dielectric and charge transport properties of polymer composites. Consequently, ORCA was selected for its strengths in probing localized molecular-scale interactions that are vital for understanding short-range behavior, while periodic DFT simulations (via VASP) were employed to capture the complementary long-range phenomena. This dual-method approach ensures a comprehensive understanding of both local and extended structural characteristics within the polymer composite system."

We believe this addition strengthens the manuscript by providing a clearer rationale for our methodological choices and addressing the limitations inherent to each computational approach. Thank you again for guiding us towards a more complete and scientifically rigorous presentation of our work.

4. Authors' response letter (559842_1_rebuttal_10311778_srtlfd.pdf):

On pages 34-35 the authors have responded extremely well and spot on to my questions by stating correctly:

"In response to the reviewer's insightful suggestion, we have expanded our computational analysis by conducting additional molecular dynamics (MD) simulations using the Vienna Ab initio Simulation Package (VASP). The implementation of periodic boundary conditions and the supercell approach within VASP enables a more accurate and holistic representation of the polymer composite system, capturing long-range electronic interactions and structural dynamics. These simulations provide deeper insights into the effects of polymer chain cross-linking on charge distribution and structural integrity, offering a more realistic depiction of the amorphous polymer matrix, including density fluctuations and packing effects. The revised manuscript now includes a detailed discussion of the VASP-based MD simulations, outlining the system configuration, computational parameters, and a comparative evaluation with the previously employed ORCA-based simulations."

However, NOTHING along these lines have been discussed in the revised manuscript. Furthermore, NO VASP MD calculations the authors state to have carried out seem to exist in the revised manuscript (559842_1_art_file_10323768_srbvtg.pdf).

The authors are requested to include this important methodological discussion also in their revised manuscript, preferably within the Section "Simulation". Furthermore, the authors are requested to display, at least some of their VASP MD computed results and compare these on their ORCA molecular modeling based MD simulations in their revised manuscript.

Response: We sincerely thank you for your thorough and constructive feedback, which has been invaluable in enhancing the clarity and accuracy of our manuscript. We regret the oversight regarding the omission of the discussion on the VASP-based MD simulations in the revised version, despite our prior communication. We greatly appreciate your careful review and the opportunity to rectify this issue.

In response to reviewer's comments, we have revised the "Simulation" section of the manuscript to include a detailed methodological discussion on the VASP-based molecular dynamics (MD) simulations. This section now includes a clear description of the computational parameters and system configurations employed in the VASP simulations, emphasizing the role of periodic boundary conditions and the supercell approach. The VASP-based MD simulations were indeed performed to capture long-range interactions and structural dynamics, such as polymer chain cross-linking, which are essential for understanding the dielectric and charge transport properties of the polymer composites.

The revised section now reads as follows: "We utilized the Vienna Ab Initio Simulation Package (VASP) to conduct molecular dynamics (MD) simulations on a polymer composite system. These simulations were performed under periodic boundary conditions, enabling the modeling of the system in three dimensions and the capture of long-range interactions, such as chain cross-linking and the effects of polymer chain packing on electronic properties. The system was modeled within a simulation box of $100 \times 100 \times 100$ Å, and a buffer distance of 20 Å between atomic configurations and the

box boundaries was maintained to mitigate finite-size effects and ensure adequate real-space separation. Electronic exchange-correlation interactions were described using the Perdew-Burke-Ernzerhof (PBE) functional within the generalized gradient approximation (GGA). A projector augmented wave (PAW) pseudopotential plane-wave basis set was employed to accurately model long-range interactions and delocalized charge distributions. Dispersion forces were included via the DFT-D3 correction scheme, which accounts for van der Waals interactions. The plane-wave energy cutoff was set to 500 eV to ensure a robust representation of the wavefunction, with an energy convergence criterion of 10^{-6} eV, ensuring high numerical precision. To complement these simulations, we compared the results with those from previous ORCA-based molecular modeling studies. ORCA, employing a quantum mechanics/molecular mechanics (QM/MM) hybrid approach, facilitated a more localized examination of molecular interactions, providing insights into short-range effects. The consistent trends observed between the results from both VASP and ORCA simulations serve to reinforce the reliability and robustness of the computational methodologies employed, thereby further validating the conclusions derived from the simulations.”

We believe that these additions address your concerns and significantly improve the clarity and completeness of the manuscript. We are confident that the revised manuscript now presents a more comprehensive and accurate description of the computational strategies employed and the corresponding results. Once again, we thank you for your insightful suggestions, which have been invaluable in strengthening the scientific rigor of our work. We look forward to your feedback on these revisions.

Figure S10. Analysis of Interaction Energy Fluctuations in Pristine PEI and PEI/DADQ Composites via VASP.

5. Authors' response letter (559842_1_rebuttal_10311778_srtlfd.pdf):

On page 35 the authors write:

"A comparative analysis of the simulation results obtained from ORCA and VASP has been incorporated into the revised manuscript to assess their respective contributions and ensure the robustness of the findings."

However, NOTHING along these computational results have been discussed or shown in the revised manuscript. (559842_1_art_file_10323768_srbvtg.pdf).

The authors are requested to discuss, even briefly, these ORCA and VASP MD simulation results in their revised manuscript.

Response: We sincerely thank the reviewer for their thorough evaluation of the manuscript and for pointing out the absence of a comparative analysis between the ORCA and VASP MD simulation results. We regret the oversight and apologize for any confusion it may have caused.

In response to this valuable feedback, we have revised the manuscript to include a comprehensive comparison of the simulation results obtained from both ORCA and VASP. This revised section details how the ORCA simulations, employing a QM/MM hybrid approach, offer critical insights into localized molecular interactions, such as dipole-dipole interactions and short-range structural fluctuations. In contrast, the VASP simulations, which utilize periodic boundary conditions, are particularly adept at capturing long-range structural features, including chain cross-linking and packing effects within the polymer matrix-factors that are vital for understanding the dielectric and charge transport properties.

Additionally, we have presented the interaction energy fluctuations in pristine PEI and PEI/DADQ composites derived from VASP MD simulations. The trends and values of the interaction energy obtained from both ORCA and VASP methods are in close agreement, further validating the reliability and robustness of the computational approaches employed in this study.

Figure S10. Analysis of Interaction Energy Fluctuations in pristine PEI and PEI/DADQ composites via (a) VASP MD Simulations and (b) ORCA MD Simulations.

We believe that this revision substantially strengthens the manuscript by offering a more thorough and detailed discussion of the simulation results. We would like to express our gratitude once again to the reviewer for their insightful comments, which have been instrumental in enhancing the clarity and scientific rigor of our work.

6. Section "Results and discussions / Modulation of energetic disorder":

On page 5 of the revised manuscript (559842_1_art_file_10323768_srbvtg.pdf) the authors write:

"As demonstrated in the Nano-IR images of the polymers (Fig.1g), the notable difference is the detection of absorption at 2140 cm⁻¹ (C≡N stretch) at the PEI/DADQ compared to pristine PEI, indicating that the DADQ molecule exhibits uniform dispersion within the polymer matrix."

In viewing the possible electronic structural and molecular conformational surface sensitivities on the PEI/DADQ polymer composite film surface I wonder how surface sensitive the Nano-IR spectroscopy will be? In other words, how surface sensitive the vibrational excitation frequencies of the DADQ molecule, such as $\nu = 2140 \text{ cm}^{-1}$, could be and why in the context of the PEI/DADQ polymer film?

The authors are advised to comment on this point also in their revised manuscript.

Response: We sincerely appreciate the reviewer's thoughtful inquiry into the surface sensitivity of Nano-IR spectroscopy, particularly regarding the vibrational excitation at 2140 cm⁻¹ (C≡N stretching mode) observed in the PEI/DADQ composite film. This is indeed an essential point to clarify, given the nanoscale nature of our analysis and the implications for molecular dispersion and film uniformity.

Nano-IR spectroscopy, employing the photothermal expansion-based approach implemented in this study, achieves spatial resolution of approximately 10 nm with exceptional surface sensitivity. The signal predominantly originates from the uppermost layers of the material, where localized thermal expansion, induced by infrared absorption, is detected via the atomic force microscope (AFM) tip. Depth sensitivity is typically restricted to the first few tens of nanometers, governed by the thermal and optical properties of the sample, enabling high-resolution characterization of molecular distributions within the near-surface region.

In the PEI/DADQ composite, the emergence of a distinct absorption band at 2140 cm⁻¹ signifies that DADQ molecules are distributed not only on the surface but also throughout the near-surface region of the polymer matrix. The spatial uniformity of this signal across Nano-IR images further supports the notion of homogeneous molecular dispersion at the nanoscale, with no detectable signs of phase separation or aggregation. Additionally, the consistent vibrational frequency at 2140 cm⁻¹ across all scanned regions indicates a stable local chemical environment surrounding the DADQ molecules. This invariance suggests that potential surface interactions-such as hydrogen bonding with PEI or minor conformational adjustments-exert minimal

influence on the vibrational characteristics of DADQ, reinforcing the conclusion of a uniformly dispersed molecular distribution within the polymer matrix.

In response to the reviewer's comment, we have incorporated the following sentence into the manuscript: “The photothermal expansion-based Nano-IR approach employed in this study achieves a spatial resolution of approximately 10 nm, offering remarkable surface sensitivity.”